METHODS

# Cell type-specific weighting-factors to solve solid organs-specific limitations of single cell RNA-sequencing

Kengo Tejima [1,2,3,4☯], Satoshi Kozawa [1,2,3,4☯], Thomas N. Sato [1,2,3,4] *

1 Karydo TherapeutiX, Inc., Kyoto, Japan, 2 ERATO Sato Live Bio-Forecasting Project, Kyoto, Japan, 3 The Thomas N. Sato BioMEC-X Laboratories, Advanced Telecommunications Research Institute International, Kyoto, Japan, 4 V-iClinix Laboratory, Nara Medical University, Nara, Japan

☯ These authors contributed equally to this work.
* island1005@gmail.com

**Data Availability Statement:** The public datasets used in this study are as follows: iOrgans Atlas http://i-organs.atr.jp Tabula Muris https://figshare.com/projects/Tabula_Muris_Transcriptomic_characterization_of_20_organs_and_tissues_

## Abstract

While single-cell RNA-sequencing (scRNA-seq) is a popular method to analyze gene expression and cellular composition at single-cell resolution, it harbors shortcomings: The failure to account for cell-to-cell variations of transcriptome-size (i.e., the total number of transcripts per cell) and also cell dissociation/processing-induced cryptic gene expression. This is particularly a problem when analyzing highly heterogeneous solid tissues/organs, which requires cell dissociation for the analysis. As a result, there exists a discrepancy between bulk RNA-seq result and virtually reconstituted bulk RNA-seq result using its composite scRNA-seq data. To fix this problem, we propose a computationally calculated coefficient, "cell type-specific weighting-factor (cWF)". Here, we introduce a concept and a method of its computation and report cWFs for 76 cell-types across 10 solid organs. Their fidelity is validated by more accurate reconstitution and deconvolution of bulk RNA-seq data of diverse solid organs using the scRNA-seq data and the cWFs of their composite cells. Furthermore, we also show that cWFs effectively predict aging-progression, implicating their diagnostic applications and also their association with aging mechanism. Our study provides an important method to solve critical limitations of scRNA-seq analysis of complex solid tissues/organs. Furthermore, our findings suggest a diagnostic utility and biological significance of cWFs.

## Author summary

Single cell RNA sequencing (scRNA-seq) is a powerful method to unveil gene expression landscape with single-cell resolution. However, scRNA-seq, in particular for the analysis of highly heterogeneous solid organs, fails to account for the apparent heterogeneity of cellular RNA contents across different cell-types. In addition, the cell dissociation-induced cryptic gene-expression is often problematic. To overcome such shortcomings, herein, we describe a concept of "cell type-specific weighting-factor (cWF)" and a computational method to calculate cWFs of diverse-cell types using intact (i.e., without cell dissociation)

from_Mus_musculus_at_single_cell_resolution/ 27733 Mouse Cell Atlas https://figshare.com/ articles/dataset/MCA_DGE_Data/5435866 scRNA-seq of skin from 9 weeks-old female C57BL6/JN mice https://www.ncbi.nlm.nih.gov/geo/query/acc. cgi?acc=GSE129218 Tabula Muris Senis (Bulk RNA-seq) https://figshare.com/projects/The_ murine_transcriptome_reveals_global_aging_ nodes_with_organ-specific_phase_and_amplitude/ 65126 Tabula Muris Senis (scRNA-seq) https:// figshare.com/projects/Tabula_Muris_Senis/64982 Human PBMC Bulk RNA-seq from healthy donor https://www.ncbi.nlm.nih.gov/geo/query/acc.cgi? acc=GSM2871599 Human PBMC scRNA-seq from healthy donor https://www.ncbi.nlm.nih.gov/geo/ query/acc.cgi?acc=GSM4557334 HEK293T/Jurkat mixture bulk RNA-seq https://www.ncbi.nlm.nih. gov/geo/query/acc.cgi?acc=GSE129240 HEK293T/ Jurkat scRNA-seq https://www.10xgenomics.com/ datasets/50-percent-50-percent-jurkat-293-t-cell-mixture-1-standard-1-1-0 The raw data (S1-S23 Tables) and the code for our algorithm are available in this paper and at https://github.com/k-teji/ VscRNAseq, respectively.

**Funding:** This work was funded by Innovative Science and Technology Initiative for Security Grant Number JPJ004596 ATLA Japan (T.N.S.), JST ERATO Grant Number JPMJER1303 Japan (T. N.S.), Nakatani Foundation (T.N.S.). K.T., S.K., and T.N.S. received salaries from Innovative Science and Technology Initiative for Security Grant Number JPJ004596 ATLA Japan, JST ERATO Grant Number JPMJER1303 Japan, and Nakatani Foundation. The funders had no role in study design, data collection and analysis, decision to publish, or preparation of the manuscript.

**Competing interests:** I have read the journal's policy and the authors of this manuscript have the following competing interests: T.N.S. is the inventor of the patents filed on the cell type-specific weighting-factors and holds shares of Karydo TherapeutiX, Inc., which owns the right to this patent. K.T. and S.K. are also employed by Karydo TherapeutiX, Inc.

whole-organ RNA-seq. Importantly, we show that cWFs are necessary for the accurate reconstitution of the whole-organ RNA-seq data using their composite scRNA-seq data and also deconvolution of the whole-organ RNA-seq data into their composite scRNA-seq data. We also show that cWFs quantitatively reflect the experimentally determined differential cellular RNA contents. These benchmarks demonstrate that cWFs indeed represent differential cellular RNA contents and/or offset the cell dissociation-induced cryptic gene-expression. Furthermore, we illustrate a medical application of cWFs by showing that the differential cWFs can effectively predict an aging-clock. In conclusion, our study reports an important methodology to solve critical limitations of scRNA-seq analysis, and also its potential diagnostic application.

## Introduction

Since the advent of single-cell RNA-sequencing (scRNA-seq) technologies early this century, this method is becoming one of the essential tools in biomedical fields [1]. While scRNA-seq provides useful information of cellular composition and gene expression at single cell resolution, it has limitations.

It is well-known that the total number of RNA molecules per cell, also known as transcriptome-size, varies from-cell-to-cell [2–4]. However, in scRNA-seq analyses, the transcripts are sequenced in individual cells and their transcript-counts are normalized, allowing the comparison of the relative abundance of each transcript-species across individual cells/cell-types [5–9]. This internal normalization process of the analyses cancels out the putative transcriptome size-variations among different cells, concealing potentially important functional differences among the cells.

This differential transcriptome-size can be determined by counting the library depth of the cells and/or by including normalizing spike-in RNA in the samples [2,3,10,11]. Alternative, it can be quantified by PCR-based methods [12,13]. However, these approaches are applicable only to culture cells, blood/immune cells, and cancer cells which are relatively uniform cell types and do not require cell dissociation. In contrast, they are not applicable to solid tissues/ organs composed of highly heterogeneous cell types. Individual cells can be dissociated from solid tissues/organs and then analyzed by scRNA-seq. However, the cell-dissociation and subsequent harvesting and/or purification steps induce cryptic gene expression [14]. Due to such limitations, it is difficult to measure and compare the absolute transcript-counts of each gene in individual cells of the complex solid tissues/organs such as the heart, brain, kidney, etc.

Furthermore, the differential transcriptome-size has biological importance. Several studies indicate that the cell-size influences the overall cellular transcription [15–18]. It is also reported that c-myc amplifies the global transcription in tumor cells [2,19], lymphocytes and embryonic stem cells [20]. Another study shows that the global transcription is repressed by the loss of MECP2, a methyl CpG binding protein, in embryonic stem cell-derived neurons [21]. The genome structure, such as ploidy/gene-dosage and nucleosome state, affects the overall transcription [12,22,23]. In additions, it is shown that the total mRNA-contents of tumor cells are associated with the extent of the cancer progression [24]. These results illustrate biological importance of accounting for the transcriptome size differences among cells.

In addition to the transcriptome-size, cell dissociation/processing-induced cryptic gene expression is another problem with scRNA-seq, in particular with solid tissues/organs [14]. The scRNA-seq analyses of the solid tissues and organs require enzymatic digesting and mechanical dissociation of individual cells. Spatial scRNA-seq requires tissue processing such

as sectioning and laser-mediated harvesting [8,9]. Such technical procedures induce cryptic gene expression [14]. The currently available scRNA-seq largely disregard this problem.

In this study, we address and overcome these limitations/problems associated with solid tissues/organs by developing a method to compute coefficients, cell type-specific weighting-factors (cWFs), to offset the transcriptome size- and cell dissociation/processing-induced problem. Furthermore, we also calculate cWFs for 76 cell-types across 10 solid organs and use them to illustrate their diagnostic and biological utilities.

## Description of the methods

### The cell type-specific weighting factor (cWF)

First, the major premise is as follows. If the transcriptome size differences and the cell dissociation/processing-induced cryptic gene expression (abbreviated as "cryptic gene expression" hereafter for convenience) are corrected in the conventional scRNA-seq, the bulk (i.e., tissue/organ) RNA-seq should be the sum of gene expression of each composite cell-type weighted by cell-type ratios.

We evaluated this premise. The synthetic whole-organ data were constructed from single-cell transcript counts combined according to the experimentally determined fraction of each cell-type in the organ. Briefly, the synthetic whole-organ RNA-seq data are calculated as the sum of the normalized transcript counts weighted by the known ratios of the composite cell-types of the organs, using the signature genes for distinguishing the cell-types for each organ (see "Datasets", "Data preprocessing", "Calculation of reference cell-type ratios", "Selection of signature gene sets" at the bottom of this section for the details). The signature genes were determined by the random forest (RF) classifier. These synthetic data for 10 organs (brain, fat, heart, kidney, liver, lung, pancreas, skin, skeletal muscle, spleen) were then compared to the corresponding real whole-organ RNA-seq data (Fig 1). The comparisons were performed by calculating their Pearson correlation coefficients.

The result shows low Pearson correlation coefficients (< 0.75) for most of the organs (brain, heart, kidney, liver, lung, pancreas, skin, skeletal muscle) regardless of the number (100, 300, 500) of the top-ranked signature genes used. Particularly low coefficients are found

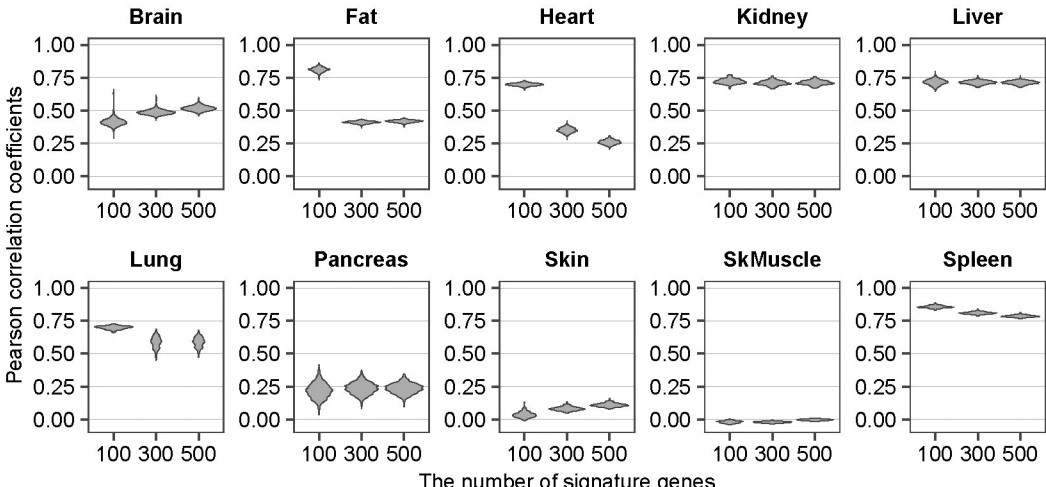

**Fig 1. Incomplete reconstitution of the whole organ RNA-seq by the composite scRNA-seq.** The similarity is shown as violin plot of the Pearson correlation coefficient for each number of the signature genes (100, 300, 500) for each organ (indicated above each plot). SkMuscle: skeletal muscle. Raw data are available as S1 Table.

with the brain ($< 0.6$ for 100, 300, 500 signature genes), the fat ($< 0.5$ for 300, 500 signature genes), the heart ($< 0.4$ for 300, 500 signature genes), the pancreas ($< 0.3$ for 100, 300, 500 signature genes), the skin ($< 0.2$ for 100, 300, 500 signature genes), and the skeletal muscle ($< 0.1$ for 100, 300, 500 signature genes). These low Pearson correlation coefficients suggest that the transcriptome-size of individual cell-types varies, resulting in the large gaps.

These results fail to satisfy the above-described major premise; hence, the differential transcriptome size-variations and/or the cryptic gene expression are unrepresented in the scRNA-seq data of the solid organs. To solve this problem, we developed an algorithm to compute such unrepresented factors (Fig 2). The concept of the algorithm is schematically described in Fig 2 and as follows: In the real scRNA-seq data analyses, the total transcript counts per cell (i.e., the transcriptome-size) of the real scRNA-seq data are normalized (i.e., the total counts per cell are equal among all composite cell-types of the organs). This results in the loss of a factor representing the putative differences of the transcriptome-size and/or the cryptic gene expression across cell-types. This factor is computed as the cell type-specific weighting-factor (cWF) (indicated as $w_1$, $w_2$, $w_3$, etc. in the Fig 2).

## Computation of cWFs

Using this algorithm, we computed cWFs for 76 cell-types across 10 organs as follows:

We calculated at least 100 cWFs for each virtual-cell/cell-subject of each cell-type and made them follow Gaussian distribution, instead of one cWF per cell-type. Based on this concept, we developed a model as follows:

$$m\mathbf{y} = \sum_j w_j \mathbf{x}_j (j \in C^m) \tag{1}$$

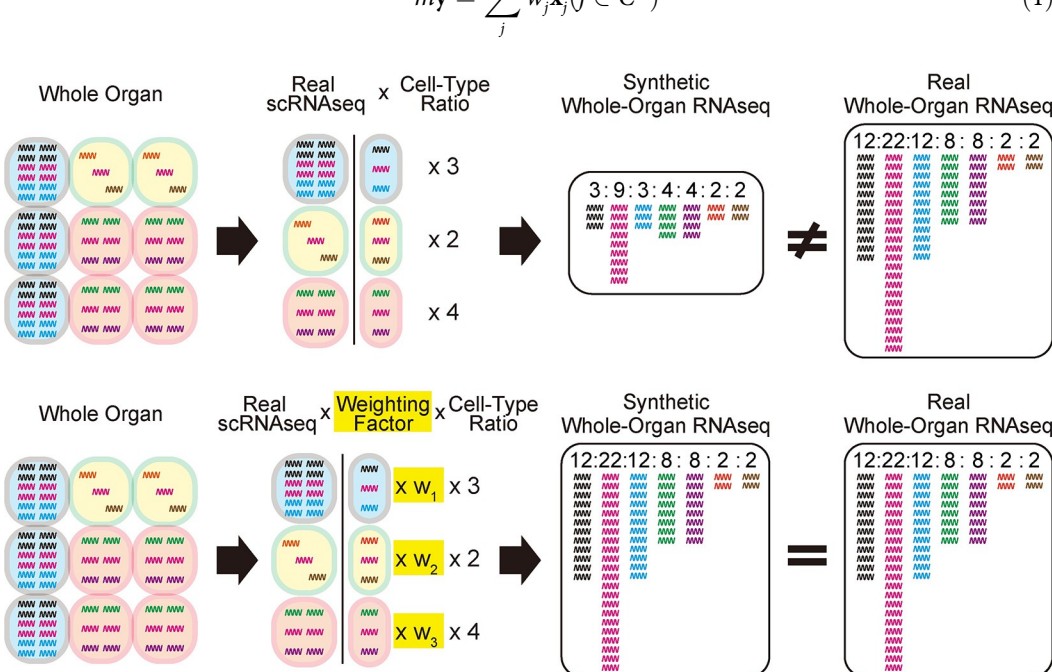

**Fig 2. Graphical description of the overall concept of the cell type-specific weighting factors (cWFs).** The top row describes the case without the cWFs, failing to reconstitute the real whole-organ RNA-seq data by the synthetic whole-organ RNA-seq data generated by assembling the composite scRNA-seq data without the cWFs. The bottom describes the case with the cWFs, accurately reconstituting the real whole-organ RNA-seq data by the synthetic whole-organ RNA-seq data generated by assembling the composite scRNA-seq data with the cWFs ($w_1$, $w_2$, $w_3$). The wiggling lines are cellular transcripts where different gene transcripts are in different colors. The ratios of the cellular transcript-counts in the synthetic and real whole-organ RNA-seq are indicated by the numbers at the top of the colored transcripts.

where $\mathbf{y} \in \mathbb{R}^n, w_j \geq 0$ and $\mathbf{x}_j \in \mathbb{R}^n$ denote the normalized whole-organ RNA-seq counts vector, the cWF for each cell subject $j$, and the normalized scRNA-seq counts vector for each cell subject $j$, respectively. The $n$ is the total number of the signature genes for the organs. In this study, 100, 300, and 500 signature genes were selected according to the ranking on the basis of 'Mean Decrease in Gini' values calculated by the RF analyses. The numbers, 100, 300, and 500 were arbitrarily selected but the smaller number of signature genes is computationally more cost-effective. The combination of cell subjects, $C^m$ was selected at random while maintaining the reference cell-type ratio described above with the total number of cell subjects, $m$, which is arbitrarily set to 100. In addition, we set $C^m$ containing at least one cell subject in each cell-type. With this model, $w_j$ was calculated by solving a quadratic problem under the constraint that the resulting value is non-negative as follows:

$$\tilde{w}_j = \operatorname{argmin}_{w_j} \sum_i^S \left| m\mathbf{y}_i - \sum_j w_j \mathbf{x}_j \right|^2 s.t. w_j \geq 0 \qquad (2)$$

Here, $S$ is the number of whole-organ RNA-seq count data. This quadratic problem was solved by using '*osqp*' package in R. The process of both random-selection of the cell subject combination and $\tilde{w}_j$ calculation was recursively performed until more than 100 $w_j$ were generated for all selected cell subjects.

The result shows significant variations of the cWFs among different cell-types in many of the organs, uncovering the body-wide degree of such variations (Fig 3). In particular, the following cell-types show significantly higher cWFs than the others within the same organ: astrocyte (AS) in the brain, endothelial cell (EC) in the fat, muscle cell-types in the heart and the skeletal muscle (CM in the heart, MC in the skeletal muscle), epithelial cell of proximal tubule (PTEC) in the kidney, epithelial cell (EP) and neuroendocrine cell (NE) in the lung, fibroblast-like cell (FB-like) in the skin, and T-cell (T) in the spleen. This result suggests that these cell-types contain larger amounts of transcripts (i.e., the larger transcriptome-size) than the others in the corresponding organs. Furthermore, the results show that such variable degrees of differences in the transcriptome-size are widespread across diverse cell-types and organs.

The default total number of cell subjects, $m$, is set to 100 in computing the cWFs. Sensitivity analysis for the total number of cell subjects was performed by varying $m$ to 50 and 200 (196 for the heart, as this is the maximal number of synthetic cell subjects allowed in the computation due to the smaller limiting number of real cardiomyocytes (CMs) in the scRNA-seq data of this organ). The results show that the overall relative pattern of the differential cWFs across the composite cell-types of each organ for the same signature gene numbers remain largely unchanged (S1 Fig, in comparison to Fig 3). However, we observed noticeable variations of the cWF values for the same cell-types with differential $m$. In particular, their values appear to decrease with the increasing $m$ for the same cell-types of the same organ. One possible explanation of this result is marginal performance of the $m$ multiplication on the left side of Eq (1) $m\mathbf{y} = \sum_j w_j \mathbf{x}_j (j \in C^m)$ where $w_j$ (cWF per cell) is calculated as a linear regression problem with the constraint of $w_j \geq 0$ (see Eq (2)). In this equation, the $m$ multiplication on the left side corrects otherwise the reduction of $w_j$ with increasing $m$. For example, when this correction function is absent (i.e., $\mathbf{y} = \sum_j w_j \mathbf{x}_j (j \in C^m)$), the increasing numbers of $\mathbf{x}_j$ are used with increasing $m$ for the fixed vector $\mathbf{y}$; hence, the $w_j$ is reduced. Therefore, it is possible that the performance of this function is marginal due to yet undefined causes.

## Datasets

The datasets for each analysis and result are summarized in S4 Table. The whole-organ RNA-seq data are 11 weeks-old male C57BL6/N Jcl mice from 'iOrgans Atlas' (http://i-organs.atr.jp)

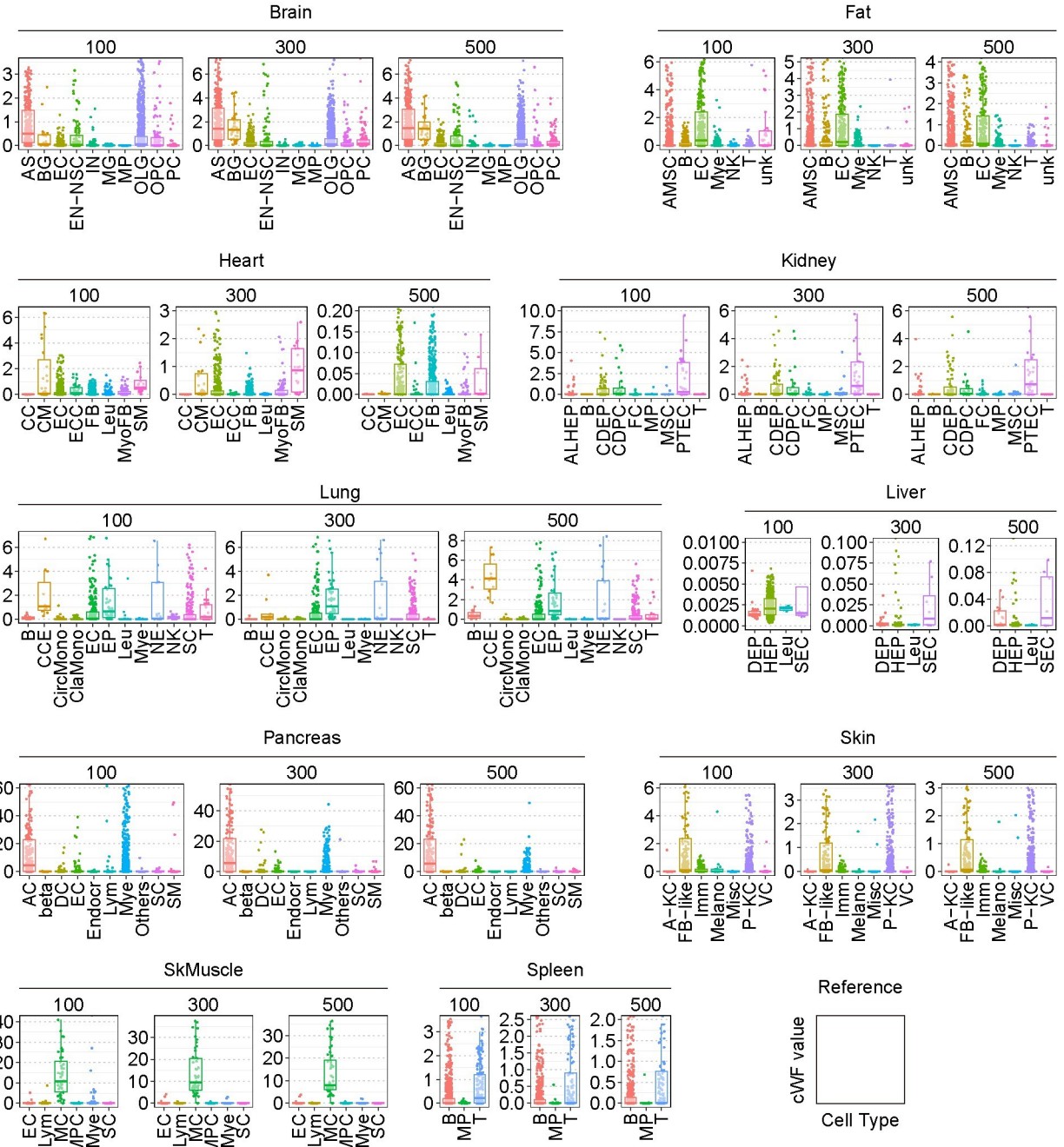

**Fig 3. The cWFs for each cell-type in each organ.** The values of the cWFs for each cell-type (bottom of each plot) for each number of the signature genes (indicated above each plot) for each organ (indicated at the top of the corresponding plots) are shown as box plots. SkMuscle: skeletal muscle. Raw data are available as S2 Table.

(deep RNA-seq data) and the same mouse organs collected and sequenced by Quant 3' mRNA-seq method [25,26], myocardial infarction (MI) model (including sham controls) CD1 male mouse organs [26], and 3, 18, and 24 months-old male C57BL6/N mice from the *Tabula Muris Senis* [27]. The Quant 3' mRNA-seq datasets are available at Gene Expression Omnibus (GEO) as GSE263816. The scRNA-seq datasets, used are the brain, the fat, the heart, the liver, the lung, and the spleen data from 11 weeks-old male C57BL6/JN mice described in *Tabula*

*Muris* [28]. The kidney data are from 3 months-old male C57BL6/JN mice in *Tabula Muris Senis* [29]. The skeletal muscle and the pancreas data are from 8–10 weeks-old female C57BL6/J reported in Mouse Cell Atlas [30]. The skin data are from 9 weeks-old female C57BL6/J mice [31]. The aging mouse scRNA-seq datasets in the brain, the heart, the kidney, the lung and the spleen are from 3, 18, 24 months-old male C57BL6/JN mice in *Tabula Muris Senis* [27,29]. The human PBMC datasets are GSM2871599 in GSE107572 (bulk RNA-seq) [43,44] and GSM4557334 in GSE150728 (scRNA-seq) [32], both are from healthy donors. The bulk RNA-seq data of HEK293T/Jurkat cells mixed at 0: 100, 20: 80, 80: 20, and 100: 0 ratios were obtained from GSE129240 as fastq files [10]. The fastq files were mapped onto the hg19 genome assembly by STAR (v2.7.7a) [33] and the count data were generated using the hg19 gene annotation by HTSeq count (v2.0.3) [34]. The scRNA-seq data of HEK293T and Jurkat cells are from 10xGenomics (https://www.10xgenomics.com/datasets/50-percent-50-percent-jurkat-293-t-cell-mixture-1-standard-1-1-0).

## Data preprocessing

Cell-type labels of the organs used in this study are as described in each corresponding original single-cell study, except for those of the skeletal muscle, the pancreas, and the heart (the *Tabula Muris Senis*, but not the *Tabula Muris*, data) and the human PBMC. For the skeletal muscle and the pancreas, the multiple labels of the cell-types of the similar gene expression patterns in the original study are combined into the single label as follows (*** indicating arbitrary characters). For the skeletal muscle, 'B cell_***', 'Dendritic cell', and 'T cell' into 'Lymphatic cell', 'Erythroblast_***', 'Granulocyte monocyte progenitor cell', 'Macrophage_***' and 'Neutrophil_***' into 'Myeloid cell', 'Muscle cell_***' into 'Muscle cell'. For the pancreas, 'B cell' and 'T cell' into 'Lymphatic cell', 'Dendritic cell', 'Erythroblast_***', 'Granulocyte', and 'Macrophage_***' into 'Myeloid cell', 'Endothelial cell_***' into 'Endothelial cell', 'Smooth muscle cell_***' into 'Smooth muscle cell', 'Stromal cell_***' into 'Stromal cell', 'Dividing cell' and 'Glial cell' into 'Others'. For the heart dataset in *Tabula Muris Senis* [29], the labels were replaced with the ones in *Tabula Muris* [28] using single-cell reference mapping by '*Seurat*' v4 [35], as follows. First, the reference scRNA-seq data in *Tabula Muris* were preprocessed by '*Seurat*' analyses with the log-normalization (where the scale factor is $10^6$), the highly variable genes selection with default parameters, the scaling for all genes, and principal component analysis (PCA) acquiring 200 principal components (PCs). The number of PCs used for the downstream analysis was determined at p-value below 0.0001 in JackStraw analysis, resulting in 33 PCs. Using these 33 PCs reference data, the query heart scRNA-seq datasets from *Tabula Muris Senis* [29] were annotated by the functions of 'FindTransferAnchors()' and 'Transfer-Data()' in '*Seurat*'. For the human PBMC, we combined cell-type labels as follows: 'Class-switched B' and 'B' into 'B', 'CD8m T', 'CD4m T', 'CD4n T', 'CD4 T', 'gd T' and 'CD8eff T' into 'T', 'IgG PB' and 'IgA PB' into 'PB', 'pDC' and 'DC' into 'DC', 'RBC', 'Platelet', 'Activated Granulocyte' and 'SC & Eosinophil' into 'Others'. For the HEK293T/Jurkat dataset, we first used the *k* = 2 k-means clustering result from 10xGenomics (https://www.10xgenomics.com/datasets/50-percent-50-percent-jurkat-293-t-cell-mixture-1-standard-1-1-0), and the cluster 1 and 2 were annotated as HEK293T and Jurkat, respectively, by the cluster marker genes. For the mouse datasets, the gene symbol-matching between the scRNA-seq and the whole-organ RNA-seq data was conducted by using entrez gene IDs derived from 'org.Mm.egALIAS2EG' in an R package, '*org.Mm.eg.db*'. ERCC-labelled genes were removed as they are spike-in genes. In addition, the counts of the three genes (*Rn45s*, *Akap5*, *Lrrc17*) were removed as they are non-mRNA artifacts significantly influencing the total counts. The normalization was performed by adjusting the total counts of each cell in the scRNA-seq datasets to a million. The

same normalization process was also applied to each of the whole-organ RNA-seq data. For the human PBMC dataset, the common genes between the scRNA-seq and the whole-organ RNA-seq were used. In the case of the human HEK293T/Jurkat dataset, we selected and used the genes linearly correlated between the scRNA-seq and the bulk RNA-seq data to offset the mismatches caused by their differential sequencing methods. Briefly, duplicate HEK293T count data (100% HEK293T bulk RNA-seq and HEK293T scRNA-seq data) were normalized to count per million and averaged. These average counts were evaluated by the Pearson correlation coefficients and fitted to the linear model with zero-interception by '*lm*' function in R. From the fitted model, the residuals were extracted and the mean and the standard deviation of the residuals were calculated. We then removed the outlier genes, determined as out of 3 times standard deviations from the mean. These processes of the Pearson correlation coefficients evaluation, the linear model fitting and the removal of the outlier genes were repeated until the Pearson correlation coefficient was over 0.9. The same processing was performed for the Jurkat cell dataset. Then, we further selected the common genes between the selected HEK293T and Jurkat gene sets, resulting in 13,135 genes (removing 6,332 from 19,467). For the 13,135 genes, we normalized all bulk and scRNA-seq count data to count per million. These preprocessed datasets were used for further analyses as described below.

## Calculation of reference cell-type ratios

The reference cell-type ratios were as described in each original single-cell study, except for the mouse heart, the mouse brain, and the human PBMC and HEK293T/Jurkat cells. For the mouse heart, the cell-type ratios described in *Tabula Muris* [28] were based on the separate analysis of the cardiac muscles and the non-muscle cells, resulting in a large under-estimation of the cardiac muscle ratio–3.1% [28]. In contrast, the ratio determined by multiple other methods and considered as the gold-standard in the cardiovascular fields is 30–40% [36–42]. According to this gold-standard ratios, we made the following modifications: We set the ratio of the cardiac-muscle cells at 30% and the remaining at 70%. The latter, then, was divided by the non-muscle cell types by maintaining their ratios the same to those in the *Tabula Muris* data. For the brain, the cell-type ratio reported in the NIH database (https://www.nervenet. org/papers/brainrev99.html#Numbers) was used to modify those in the *Tabula Muris* scRNA-seq data [28], as follows. First, we divided the brain cell-type classes into four classes: 'neurons', 'glial cells', 'endothelial cells', 'others', and then set the ratio of these classes at 75: 23: 7: 4, respectively, according to their estimated ratios in the NIH murine brain database (https:// www.nervenet.org/papers/brainrev99.html#Numbers). Second, we further divided 'neurons', 'glial cells', and 'others' into more cell-type classes according to *Tabula Muris* [28]. The 'neurons' were further divided into 'neuron-excitatory neurons and some neuronal stem cells' and 'neuron-inhibitory neurons'. The 'others' class was divided into 'brain pericyte-NA' and 'oligodendrocyte precursor cell-NA'. For the 'glial cells' class, we introduced three assumptions: 1) the 'glial cells' can be classified into four cell-types, 'microglial cell-NA', 'astrocyte-NA', 'Bergmann glial cells-NA', and 'oligodendrocyte-NA', according to *Tabula Muris* [28]; 2) the ratios of these four glial cell types follow those of *Tabula Muris* [28] among them, and 3) the ratio of 'microglial cells-NA' in the whole brain is 0.1, as it is reported that the 'microglial cells' account for 10–15% of the whole brain cells. On the basis of these assumptions, the ratios of each brain cell type was estimated as 'macrophage-NA' (ca. 0.2%), 'microglial cell-NA' (10.0%), 'astrocyte-NA' (ca. 2.2%), 'Bergmann glial cell-NA' (ca. 2.1%), 'brain pericyte-NA' (ca. 1.5%), 'endothelial cell-NA' (ca. 6.4%), 'neuron-excitatory neurons and some neuronal stem cells' (ca. 47.5%), 'neuron-inhibitory neurons' (ca. 21.3%), oligodendrocyte-NA' (ca. 8.7%) and 'oligodendrocyte precursor cell-NA' (ca. 1.9%) and used as the reference. For the human PBMC, the

reference cell-type ratios are as described [43,44], except for those of "CD16$^+$ monocytes" and "PB and Others". The ratios of these cell-types are derived as follows: The ratios of the cell-type label "the others" was divided to 'CD16$^+$ monocytes' (ca. 7.21%), 'PB' (ca. 0.24%), 'Others' (ca. 3.77%), according to their RNA-seq data-based ratios.

## Selection of signature gene sets

Random Forest (RF) was performed to extract signature genes for distinguishing cell-types from one another in each organ using the scRNA-seq count data derived from the cell-type reference datasets described in the above section. In this study, the '*randomForest*' package in R was used for tuning and producing a classifier by RF. The scRNA-seq data were first divided by 8:2 into the training and the test data. Using the training data, we determined two parameters, 'mtry' and 'ntree' for producing an RF-model. The parameter 'mtry' means the number of features used for an RF-model and was tuned by a function 'tuneRF()', while the parameter 'ntree' means the number of trees generated in an RF-model and was set as 500, which is sufficient to converge error rate in cell-type classification. The produced RF-model was validated by using the test data and F1-score calculated by a function 'F1_Score()' in an R package '*MLmetrics*'. Then we confirmed the validity of the model by F1-score over 0.8. Following the classifier production, the important features in the classifier were extracted as the signature genes for each cell-type, where we used 'Mean Decrease in Gini' values as the importance indicator for each gene.

## Verification and comparison

We validate the computed cWFs by five methods: 1) Reconstitution (Figs 4 and S2–S5), 2) deconvolution (Figs 5 and 6) of the whole-organ RNAseq data using their composite scRNA-seq data, 3) the comparison of the cWFs to the experimentally measured transcript-contents (Figs 7 and S6), 4) independence of cWFs from differential cell-type ratios (S7 Fig), and 5) prediction using unmatched organ RNA-seq and scRNA-seq data (S8 Fig).

## Validation by reconstitution

First, we validated the cWFs by the reconstitution (Fig 4) (see also "Datasets", "Data preprocessing", "Calculation of reference cell-type ratios", "Selection of signature gene sets" in the

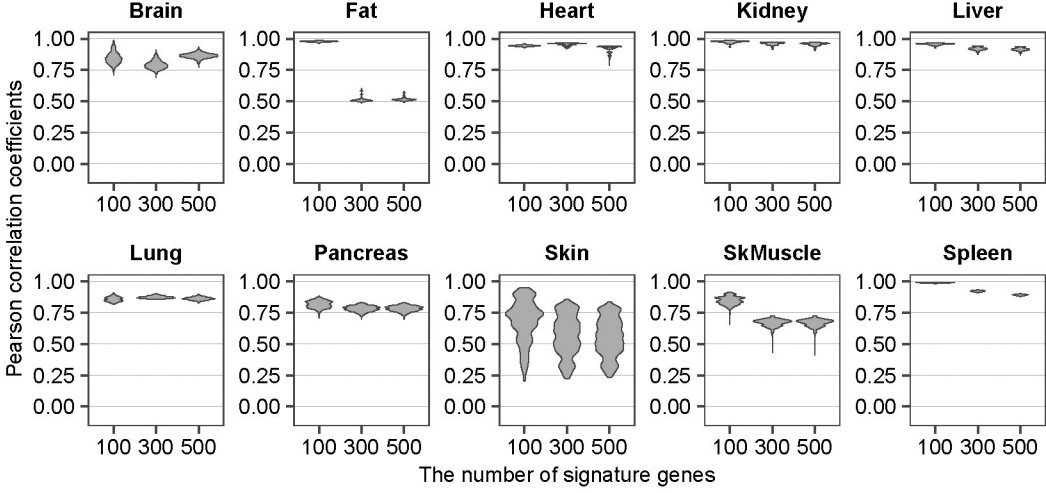

**Fig 4. Accurate reconstitution of the whole organ RNA-seq by the composite scRNA-seq and cWFs.** The similarity is shown as violin plot of the Pearson correlation coefficient for each number of the signature genes (100, 300, 500) for each organ (indicated above each plot). SkMuscle: skeletal muscle. Raw data are available as S5 Table.

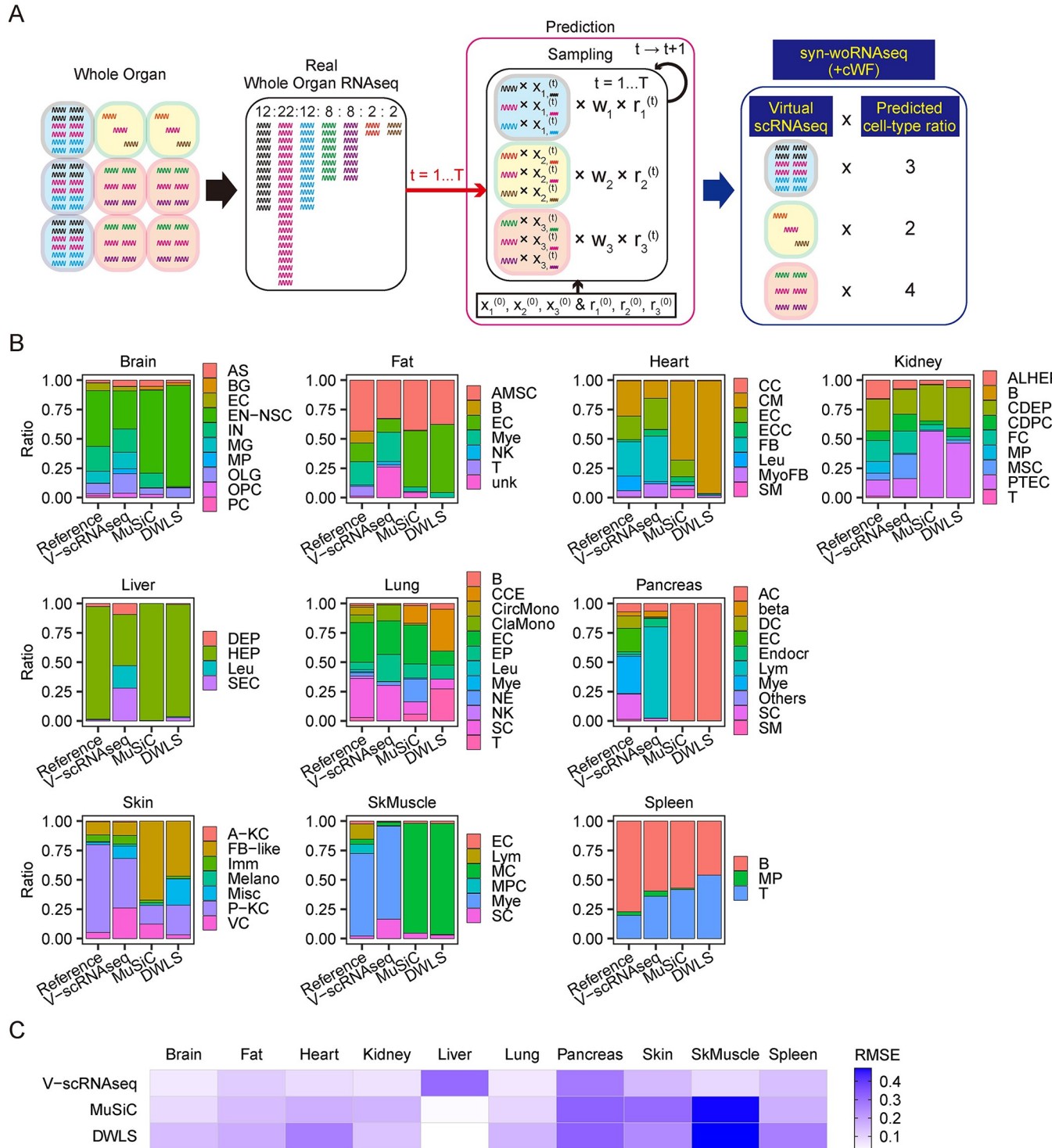

**Fig 5. Accurate deconvolution of the whole organ RNA-seq by cWFs. (A)** Graphical description of the overall concept of the deconvolution method with the cWFs. Shown is the case of an organ consisting of 3 cell-types (1, 2, 3) with 7 different genes (wiggling lines of different colors). The distribution of the cWFs: $w_1, w_2, w_3$ for each cell-type 1, 2 and 3, respectively. The virtual transcript counts ($x_1^{(t)}, x_2^{(t)}, x_3^{(t)}$ where the different genes are indicated by the wiggling lines of different colors for each cell-type) and the ratio ($r_1^{(t)}, r_2^{(t)}, r_3^{(t)}$) of virtual cell-type 1, 2 and 3 at each iteration, t, are as indicated. Their initial conditions are $x_1^{(0)}, x_2^{(0)}, x_3^{(0)}, r_1^{(0)}, r_2^{(0)}, r_3^{(0)}$. The iteration (t = 1, ..., T) is performed until both **X** and **r** converge. Upon completing the iterations, the sum of the virtual transcript counts weighted by the cWFs (indicated as "Virtual scRNAseq") weighted by the virtual cell-type ratios (indicated as "Predicted cell-type ratio") generates the synthetic whole-organ transcriptome (indicated as "syn-woRNAseq (+cWF)"), as shown at the far right of this panel. **(B)** Bar graph showing the cell type-ratios computed by our deconvolution method (V-scRNAseq) for each organ. The deconvolution was performed with the cWFs computed using the

optimal number of the signature genes for each organ. The reference cell-type ratios and the ratios estimated by the conventional deconvolution methods without cWFs, MuSiC and DWLS, are shown as comparisons. The bar graphs are composed of the cell-types computed to be present for each organ by our method. **(C)** The quantitative similarity of the computed ratios to the reference ratios is shown by heatmap of RMSE for our method with cWFs (V-scRNAseq), MuSiC, and DWLS. RMSE: Root Mean Squared Errors. SkMuscle: skeletal muscle. Raw data for Fig 5B and 5C are available as S11 Table.

previous Description of the Methods section for the details). Using the individual cWFs for each cell-type (i.e., one cWF for each cell subject up to the total number of cell subjects, $m$), we weighted the transcript counts for each cell-type. We then summed these weighted transcript counts for all composite cell-types of each organ according to their ratios to construct the synthetic whole-organ RNA-seq (syn-woRNAseq(+cWF)), data. Then, they are compared to the corresponding real whole-organ RNA-seq (r-woRNAseq) data. Specifically, in every recursion of computing the cWFs as described in "Computation of cWFs" of the Description of Methods section, Pearson correlation coefficients were calculated between the real whole-organ RNA-seq and the corresponding synthetic whole-organ RNA-seq generated with or without the weighting-factors (Figs 1 and 4). The result shows the Pearson correlation coefficients of 0.8–1.0 between the syn-woRNAseq(+cWF) and r-woRNAseq for all 10 organs with all or at least one of the 100, 300, 500 signature genes (Fig 4). These are significant improvements from the 0–0.75 of the Pearson correlation coefficients without the cWFs (Fig 1). The heart shows the Pearson correlation coefficients of almost 1.0 with the cWFs (Fig 4), as compared to those of 0.25–0.7 without the cWFs (Fig 1). Even for the spleen where the Pearson correlation coefficients are 0.75–0.8 without the cWF (Fig 1), the cWF (i.e., syn-woRNAseq(+cWF)) improves the Pearson correlation coefficients up to 0.9–1.0 (Fig 4). The results demonstrate that cWFs correct the large gap between the synthetic whole-organ RNAseq without the cWFs (syn-woRNAseq(-cWF)) and r-woRNAseq, supporting the notion that cWFs reflect the transcriptome size-variations of the cells and their requirement for more accurate single-cell analyses. Despite the variations of the cWF values induced by the differential $m$ (Figs 3 and S1), the reconstitution with these varying cWFs shows similarly significant improvement to that without cWFs (S2 Fig, compare to Fig 1). One noticeable difference is that the increase of $m$ appears to slightly improve the Pearson correlation coefficients (S2 Fig), suggesting a benefit of increasing $m$. The similar improvement was found with additional organ RNA-seq datasets (S3 Fig), further supporting the notion that the method does not necessarily require both whole-organ RNA-seq and the corresponding scRNA-seq data at the same time. In fact, the relative differences of cWFs among the composite cell-types of each organ remained similar regardless of the organ RNA-seq datasets (Figs 3, S4, and S5). Only exception is the liver, where the relative differences of cWFs among the liver cell-types appear variable. The liver is predominantly hepatocyte (>95%), suggesting that the method is effective with highly-heterogeneous organs, but not with relatively homogenous organs such as the liver.

## Validation by deconvolution

Next, the validation by the deconvolution (Fig 5) was performed (see also "Datasets", "Data preprocessing", "Calculation of reference cell-type ratios", "Selection of signature gene sets" in the previous Description of the Methods section for the details). We developed a deconvolution method that integrates cWFs in a Bayesian framework, as schematically described in Fig 5A. It works as follows: The distribution of the cWFs ($w_1$, $w_2$, $w_3$ in Fig 5A, the lower case 1, 2, 3 indicate cell-type 1, 2, 3) is used to compute virtual transcript counts ("x" in Fig 5A) for each gene (indicated as wiggling lines of different colors in Fig 5A) and the ratio (indicated as "r" in Fig 5A) of each cell-type in each organ (painted in different colors in Fig 5A). Using the signature-gene set, the computation is iteratively performed by the sampling method of No-U-Turn

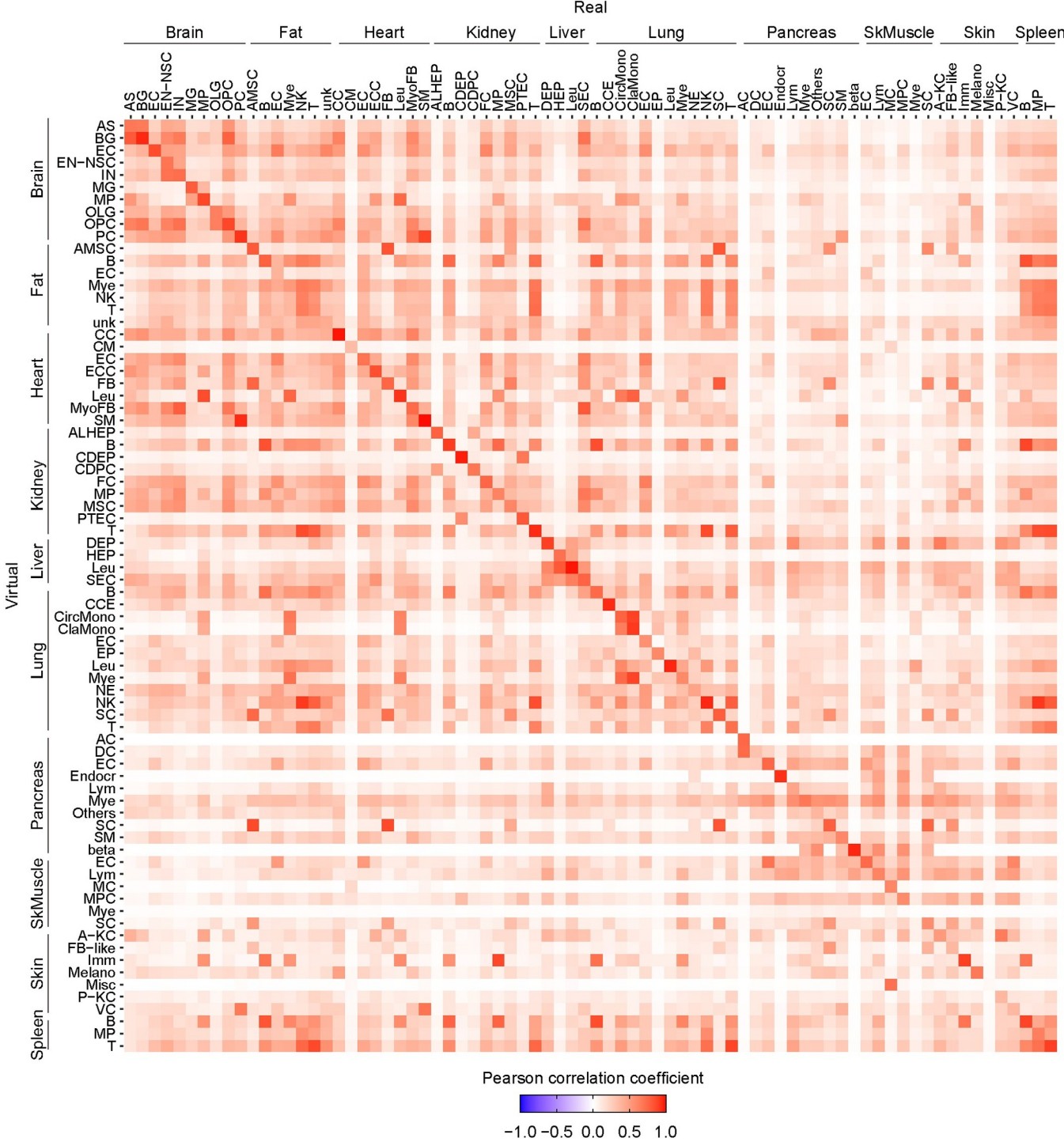

**Fig 6. Computation of the complete transcriptome of the composite cell-types of 10 mouse organs.** The heatmap of Pearson correlation coefficients for the transcriptome of all virtual and real cell-types across all 10 organs. The Pearson correlation coefficients are calculated for 23,131 (Brain, Fat, Heart, Lung, Spleen), 22,742 (Kidney), 23,104 (Liver), 15,682 (Pancreas), 21,233 (Skin) and 14,129 (Skeletal muscle) genes. Raw data are available as S12 Table.

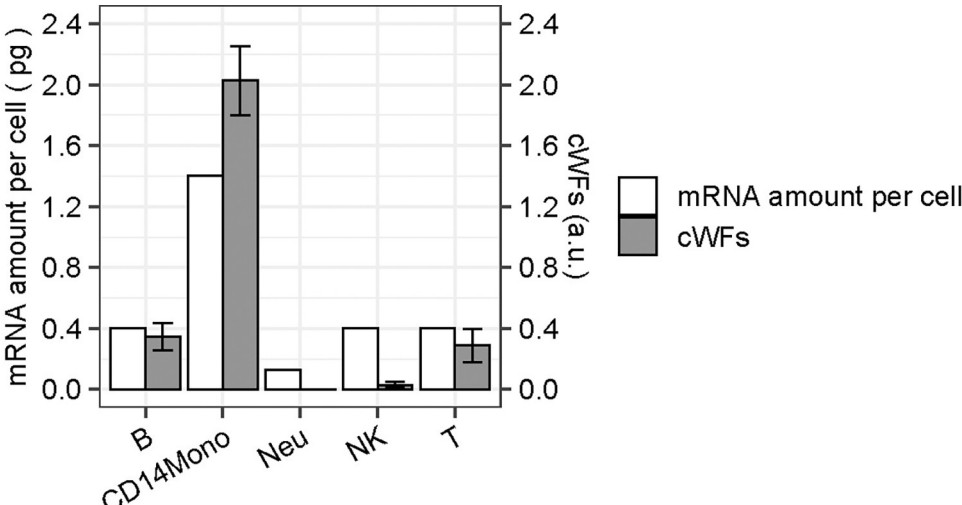

**Fig 7. Comparison of the cWFs to the experimentally-determined total mRNA-contents of human PBMC cell-types.** The comparison is shown as bar graphs of the experimentally-determined mRNA-amounts per cell (mRNA amount per cell: empty bar) and the computed cWFs (cWFs: filled-bar) for each cell-type. The bars are shown as mean ± S.E. (standard error). B: B-cell, CD14Mono: CD14[+] monocytes, Neu: Neutrophil, NK: Natural killer cell, T: T-cell. The validation of the cWFs by reconstitution and deconvolution are found in S6 Fig. Raw data are available as S13 Table.

Sampler (NUTS), in the Bayesian framework with two hyperparameters, α and β, to account for the combinatorial influence of the cell-type ratios and the gene expression patterns of each cell-type on the whole-organ-level transcriptome, respectively. For the iteration (t = 1, . . ., T), the initial condition is set as $x_1^{(0)}$ and $r_1^{(0)}$ for the transcript counts and the ratio of cell-type 1 (see the diagram labels as "Prediction" in Fig 5A). Those for cell-types 2 and 3 are indicated accordingly (see the diagram labels as "Prediction" in Fig 5A). Such initial conditions are the cWF-weighted transcript count for each gene and the reference cell-type ratio, respectively. The iteration is repeated until all sampling variables converge with the signature-gene set. The counts of the non-signature gene set are computed by an analytic approach in the Bayesian framework with a hyperparameter, γ, which has the similar nature of *β* as above. The "r" is fixed as the above-estimated value, and only "x" in the non-signature gene set is computed.

The specific computational operations of the deconvolution are as follows:

By assuming that the distribution of the weighting-factors follows Gaussian distribution, we calculated the mean and variance of the weighted counts for each cell subject as follows:

$$\tilde{\mathbf{x}}_j \sim \mathcal{N}\left(\mu_{w_j}\mathbf{x}_j, \sigma^2_{w_j}\mathrm{diag}(\mathbf{x}_j \odot \mathbf{x}_j)\right) \tag{3}$$

where, $\tilde{\mathbf{x}}_j$, $\mu_{w_j}$, and $\sigma^2_{w_j}$ denote the weighted count vector for the cell subject *j*, the mean of the cWFs for the cell subject *j*, and the variance of the cWFs for the cell subject *j*, respectively. The operator $\odot$ represents the element-wise product between two vectors. On the basis of the computed mean and variance of the weighting-factors for each cell subjects, the weighted count vector of cell-type *k* was calculated by assuming that they follow Gaussian distribution as follows:

$$\bar{\mathbf{x}}_k \sim \mathcal{N}\left(\frac{1}{N_k}\sum_j \mu_{w_j}\mathbf{x}_j, \frac{1}{N_k}\sum_j \sigma^2_{w_j}\mathrm{diag}(\mathbf{x}_j \odot \mathbf{x}_j)\right)(j \in C^k) \tag{4}$$

where, *k*, $C^k$, and $N_k$ denote a cell-type, a set of subjects labelled with cell-type *k*, and the

number of subjects in $C^k$, respectively. Using these weighted count vectors for the cell subjects, the model is built as follows:

$$\mathbf{y} = \mathbf{Xr} \tag{5}$$

where, $\mathbf{y}$, $\mathbf{X} = [\bar{\mathbf{x}}_1, \ldots, \bar{\mathbf{x}}_k, \ldots, \bar{\mathbf{x}}_K]$, and $\mathbf{r}$ are the whole-organ RNA-seq data vector, the matrix of the virtual transcript counts where the columns are the weighted-counts for each cell-type calculated as above, and the coefficient vector corresponding to the cell-type ratio, respectively. To compute $\mathbf{X}$ and $\mathbf{r}$, we employed Bayes' theorem. To apply Bayes' theorem, Gaussian noise $\mathcal{N}(0, \beta\mathbf{I})$ was added to Eq (5) and a probabilistic model was developed as follows:

$$P(\mathbf{y}|\mathbf{X}, \mathbf{r}) = \mathcal{N}(\mathbf{y}|\mathbf{Xr}, \beta\mathbf{I}) \tag{6}$$

where, $\beta$ denotes a hyperparameter. According to the Bayes' theorem, the posterior distribution of $\mathbf{X}$ and $\mathbf{r}$ was obtained as

$$P(\mathbf{X}, \mathbf{r}|\mathbf{y}) \propto P(\mathbf{y}|\mathbf{X}, \mathbf{r})P(\mathbf{X})P(\mathbf{r}) \tag{7}$$

where, $P(\mathbf{X})$ and $P(\mathbf{r})$ denote prior distribution of $\mathbf{X}$ and $\mathbf{r}$, respectively. $P(\mathbf{X})$ and $P(\mathbf{r})$ are given as follows:

$$P(\mathbf{X}) = \prod_{k=1}^{K} P(\bar{\mathbf{x}}_k) = \prod_{k=1}^{K} \mathcal{N}(\bar{\mathbf{x}}_k|\boldsymbol{\mu}'_k, \boldsymbol{\Sigma}_k^{-1}) \tag{8}$$

$$P(\mathbf{r}) = Dirichlet(\mathbf{r}|\alpha\mathbf{1}_K) \tag{9}$$

where, $\alpha$ and $\mathbf{1}_K$ denote a hyperparameter and the $K$-dimensional all-ones vector. In prior distribution $P(\mathbf{X})$, $\mathcal{N}(\bar{\mathbf{x}}_k|\boldsymbol{\mu}'_k, \boldsymbol{\Sigma}_k^{-1})$ is the same as Eq (4). In other words, $\boldsymbol{\mu}'_k = \frac{1}{N_k}\sum_j \mu_{w_j}\mathbf{x}_j$ and $\boldsymbol{\Sigma}_k^{-1} = \frac{1}{N_k}\sum_j \sigma_{w_j}^2 \mathrm{diag}(\mathbf{x}_j \odot \mathbf{x}_j)$. To maximize the posterior distribution, we employed an extension of Hamiltonian Monte Carlo (HMC) sampler, the No-U-Turn Sampler (NUTS) by a software, 'stan', which was performed on R program by an R package, '*rstan*', and used only signature gene sets for sampling, as the running time is reduced. The probabilities of hyperparameters, $\alpha$ and $\beta$ were set as zero-truncated standard normal distribution. In addition, we set zero for the negative values in the sampled elements of $\mathbf{X}$. The 'stan' sampling parameters, such as 'the number of iterations', 'max_treedepth', 'adapt_delta', 'thin', etc., were appropriately chosen to converge well the sampled distributions. The convergence was evaluated by R-hat values (Gelman-Rubin statistics) in all sampling variables below 1.1, which is a generally accepted value for convergence evaluation. For the expression pattern of non-signature genes, we set another noise parameter $\gamma$ in Eq (6), instead of $\beta$ and estimated $\bar{\mathbf{x}}_k$ (the expression patterns in non-signature genes) by the following probability equation:

$$P(\bar{\mathbf{x}}_k|\mathbf{y}, \bar{\mathbf{r}}, \{\bar{\mathbf{x}}_l\}_{l\neq k}) \propto P(\mathbf{y}|\mathbf{X}, \bar{\mathbf{r}})P(\bar{\mathbf{x}}_k) \tag{10}$$

where, $\bar{\mathbf{r}}$ denotes the mean values of $\mathbf{r}$ determined by the sampled distribution above.

Then, $P(\bar{\mathbf{x}}_k|\mathbf{y}, \bar{\mathbf{r}}, \{\bar{\mathbf{x}}_l\}_{l\neq k})$ follows Gaussian distribution and its mean and variance can be calculated by the following equations:

$$\mathbb{E}[\bar{\mathbf{x}}_k] = (\gamma\mathbf{r}_k^2\mathbf{I} + \boldsymbol{\Sigma}_k^{-1})^{-1}(\gamma\mathbf{r}_k\mathbf{y} + \boldsymbol{\Sigma}_k^{-1}\boldsymbol{\mu}'_k - \gamma\sum_{l\neq k}\mathbf{r}_k\mathbf{r}_l\bar{\mathbf{x}}_l) \tag{11}$$

$$\mathrm{Var}[\bar{\mathbf{x}}_k] = \gamma\mathbf{r}_k^2\mathbf{I} + \boldsymbol{\Sigma}_k^{-1} \tag{12}$$

In the resulting $\mathbb{E}[\bar{\mathbf{x}}_k]$, all negative values were set to zero. The hyperparameter $\gamma$ was set as either of $10^{-5}$, $10^{-4}$, ..., $10^5$. The estimating calculation was recursively performed until the root

mean square error (RMSE) of $\bar{\mathbf{x}}$ between iterations gets below 1 or until at most 100 times enough to converge, where one iteration is the estimating calculations of Eq (11) and (12) for all $k$ (i.e., all cell-types) and for each iteration, $\bar{\mathbf{x}}_l$ is from the prior iteration. The optimal value of the hyperparameter $\gamma$ was determined by the least RMSE between the original whole-organ RNA-seq (i.e., $\mathbf{y}$) and the estimated one (i.e., $\bar{\mathbf{y}}$). The optimal number of the signature genes was determined based on the three criteria as follows: 1) Improvement of the Pearson correlation coefficient between the r-woRNAseq and the syn-woRNAseq(+cWF), 2) Convergence in the deconvolution, 3) Higher similarity to the reference cell type composition (i.e., lower RMSE).

The deconvolution results for 10 organs using this method were compared to the real cell-composition data of the corresponding organ (Fig 5B and 5C). The results were also compared to those of two conventional deconvolution methods without the cWF, MuSiC and DWLS [45,46]. These two were chosen as they have been applied to 1–4 organs and appear to outperform the other published methods. MuSiC was conducted as described [46]. DWLS was performed as described [45], except that a function, 'solve.QP()' (R package: '*quadprog*') was substituted by 'solve_osqp()' (R package: '*osqp*') for quadratic problem solver.

The result shows that our method (V-scRNAseq) outperforms the methods without the cWFs (MuSiC, DWLS) for 9 organs (brain, fat, heart, kidney, lung, pancreas, skin, skeletal muscle, spleen). For these organs, the cell-type ratios predicted by our method are more similar to those of the real data than those predicted by MuSiC or DWLS (Fig 5B). Notably, our method corrected the abnormally large ratios of cardiac muscle and muscles in the heart and the skeletal muscle, respectively, estimated by MuSiC and DWLS (Fig 5B). The lower root-mean-squared error (RMSE) for these 9 organs with our method (Fig 5C) further confirms the better performance of our method (Fig 5C). For the liver, our method failed (Fig 5B and 5C), despite the high performance in the reconstitution (Fig 4). This may be due to the predominantly hepatocyte composition of the liver. Our method assumes the symmetric Dirichlet distribution as prior distribution which influences the posterior distribution–hence, the method is less suitable for the organ, such as the liver, consisting of a single dominant cell-type.

In addition to the cell-type ratios, the deconvolution method yields the virtual transcriptome of 23,131 (Brain, Fat, Heart, Lung, Spleen), 22,742 (Kidney), 23,104 (Liver), 15,682 (Pancreas), 21,233 (Skin) and 14,129 (Skeletal muscle) genes in each of the "virtual" cell-type. Hence, we compared these virtual transcriptome profiles to those of the corresponding real cell-type for the 10 organs (Fig 6). Our approach computes a posterior distribution and its Expected *A Posteriori* (EAP) for counts of each transcript for each cell-type. Using these EAPs, we then calculated the mean counts of each cell-type in each organ. For the virtual scRNA-seq, the estimated mean count for each cell-type was normalized to a million. For the real scRNA-seq, the mean count for each individual cell was normalized to a million and averaged for each cell-type. Using these normalized counts, we calculated Pearson correlation coefficients for each cell-type.

Their Pearson correlation coefficients indicate that the virtual transcriptomes are comparable to the real ones across all cell-types and organs. They also recapitulate the transcriptomics-relatedness of the same/related cell types across different organs (Fig 6). For example, the similarity of endothelial cells in multiple organ such as brain, fat, heart, liver and lung is recapitulated in the virtual transcriptomes among these organs (Fig 6). The relatedness of each of the immune system cells across multiple organs is also represented (Fig 6).

## Validation by comparing the cWFs to the experimentally measured transcript-contents

The reconstitution and deconvolution results demonstrate that cWFs enable the accurate representation of individual cells in the context of the whole-organs of highly heterogeneous

compositions, presumably by weighting the variations of the transcriptome-size of different cell-types. We further validate this account by directly comparing the cWFs variations to those of the experimentally-determined total mRNA-contents of the corresponding cell-types.

The measurement of such variations for individual single-cells of solid organs are difficult and the methods to measure such variations are limited to cultured cells and systemic/circulating cells such as immune cells, and tumor cells [2,10,13,24]. Therefore, we used the experimentally-determined total mRNA-contents of the cell-types composing human peripheral blood mononuclear cells (PBMCs) and compared them to the computed cWFs of the corresponding cell-types (Fig 7).

We used bulk and single-cell RNA-seq data of human PBMCs [32,43,44] and computed the cWFs of B-cells, CD14[+] monocytes, CD16[+] monocytes, dendritic cells, neutrophils, natural killer (NK) cells, plasmablasts (PB), T-cells, and the others (e.g., red-blood-cells, platelets, etc.) (see also "Datasets", "Data preprocessing", "Calculation of reference cell-type ratios", "Selection of signature gene sets" in the previous Description of the Methods section for the details). The reconstitution analysis confirmed that they correct the biased gene-expression in the scRNA-seq data caused by the normalization of the transcriptome-size, as indicated by the higher Pearson correlation coefficients (S6A Fig). The deconvolution analysis was also conducted using the optimal number of the selected signature genes. The result shows the cell-type composition that is similar to the reference as indicated by low RMSE (0.09) and high Pearson correlation coefficient (0.84) (S6B Fig).

Based on these results, we made side-by-side comparison of the computed cWFs and their experimentally-measured transcriptome-size of the corresponding cell-types (Fig 7). The total mRNA-amounts are experimentally measured and determined for B-cells, CD14[+] monocytes, neutrophils, NK cells, and T-cells of PBMCs [11]. The result shows that the computed cWFs accurately recapitulate the 3.5–10.8-fold higher transcriptome-size of the CD14[+] monocytes compared to the B-, T-, NK cells, or neutrophils, respectively (i.e., CD14[+] monocytes: 1.4 pg/cell vs. B-/T-/NK-cells: 0.4 pg/cell, Neutrophils: 0.13 pg/cell). It also recapitulates the 3-fold higher values of the B- and T-cells than that of neutrophils (i.e., 0.4 vs. 0.13 pg/cell). The experimental values of the mRNA-amounts of B- and T-cells are approximately the same, which is also recapitulated by virtually the same cWFs of these two cell-types. These side-by-side comparisons provide another layer of evidence supporting the account that cWFs indeed represent the transcriptome size-variations among different cell-types.

## Validating the independence of cWFs from differential cell-type ratio

Next, we examined whether differential cell-type ratios influence the cWFs. For this purpose, we computed the cWFs using the bulk RNA-seq data of two cell-types that are mixed to the predetermined differential ratios (S7 Fig) (see also "Datasets", "Data preprocessing", "Selection of signature gene sets" in the previous Description of the Methods section for the details). In this analysis, we used the bulk RNA-seq data derived from the samples where HEK293T and Jurkat cells are mixed to the ratios of 20: 80 and 80: 20 [10]. The RNA content of HEK293T cells is determined to be approximately six-times more than that of Jurkat cells [10]. The result shows the expected approximately six-fold larger RNA content of HEK293T cells, as compared to that of Jurkat cells, regardless of their differential ratios (i.e., 20: 80 vs. 80: 20) (S7 Fig). This result supports the notion that the cWFs are unaffected by differential cell-type ratios.

## Validation by prediction using unmatched organ RNA-seq and scRNA-seq data

We next evaluated whether our method always requires the exactly matched organ (bulk)-RNA-seq and scRNA-seq data of the organ of interest. For this purpose, we performed the deconvolution of the heart RNA-seq data of a myocardial infarction model [26,47], using the

cWFs of the normal heart (i.e., sham control shown in S5 Fig) to compute a putative change of the cell-type ratios of the infarcted heart (S8 Fig) (see also "Datasets", "Data preprocessing", "Calculation of reference cell-type ratios", "Selection of signature gene sets" in the previous Description of the Methods section for the details). The results show that the method can detect the expected cell-type ratio changes [48,49]–i.e., the significantly decreased cardiomyocytes (CM) at the early MI stage (ca. ×0.62) (S8E Fig) and their further reduction at the middle fibrosis stage (ca. ×0.42) (S8M Fig), and increased fibroblasts (FB) at the fibrosis (ca. ×1.61) (S8M Fig) and late cardiac remodeling (ca. ×1.70) (S8L Fig) stages. These results suggest that the deconvolution method herein does not necessarily require the prior knowledge about the exact cell-type ratio of the target organ.

## Applications

### Analysis of differential cWFs in aging progression

Previously, biological and diagnostic utilities of differential transcriptome size are reported [2,12,15–24]. Hence, we explored such utilities for cWFs. We hypothesized that the transcriptome-size of each cell-type differentially change over the course of aging and that such a phenotype could underlie the aging mechanism and could also be exploited as a new type of aging-biomarker. To test this hypothesis, we examined whether the patterns of the cWFs change over the course of aging. Furthermore, we examined whether such putative changes of the cWFs actually indicate the progression of aging.

For this purpose, we used the *Tabula Muris Senis*, the atlas of the multiple-organ single-cell transcriptomics across multiple aging stages of mouse [27,29] (see also "Datasets" in the previous Description of the Methods section for the details). This atlas provides both the whole-organ RNA-seq data and their corresponding composite scRNA-seq data. Using these data, we computed cWFs of the composite cell-types of 5 organs (brain, heart, kidney, lung, spleen) across 3 aging-stages (3, 18, and 24 months-old) of mice. The 3, 18, and 24 months-old (mos.) in mouse correspond to 25–26 years-old, 60–80 years-old, and $\geq$ 80 years-old in human, respectively [50]. These 5 organs were chosen for the following two reasons: 1) they performed well in both the reconstitution and deconvolution validations (Figs 4 and 5); 2) their equivalent whole-organ/single-cell data are also present in our 10 organs-analysis described above.

Using these datasets, we computed the cWFs of all composite cell-types for each organ (see also "Data preprocessing", "Calculation of reference cell-type ratios", "Selection of signature gene sets" in the previous Description of the Methods section for the details). The reconstitution (S9 Fig) and deconvolution (S21 Table) analyses confirmed that they correct the biased representations of the gene expression and cell-type compositions, respectively, in the scRNA-seq data with one or more of the signature gene set(s) for each aging-stage. Based on these results, we selected the best performing numbers of the signature genes for each organ and aging-stage by omitting the conditions that failed to significantly improve the reconstitution (highlighted in pink in S21 Table) and selecting the ones with the lowest RMSE and the highest Pearson correlation coefficients in the deconvolution analysis (highlighted in light-green in S21 Table). This screening selected the cWFs computed with the following conditions: Brain (3 mos.– 500 signature genes, 18 mos.– 300 signature genes, 24 mos.– 300 signature genes), Heart (3 mos.– 100 signature genes, 18 mos.– 500 signature genes, 24 mos.– 300 signature genes), Kidney (3 mos.– 300 signature genes, 18 mos.– 500 signature genes, 24 mos.– 500 signature genes), Lung (3 mos.– 500 signature genes, 18 mos.– 300 signature genes, 24 mos.– 300 signature genes), Spleen (3 mos.– 300 signature genes, 18 mos.– 100 signature genes, 24 mos.– 500 signature genes).

With these selected cWFs, we examined whether their relative differences change over the course of aging. For this purpose, we compared the ratios of the cWFs of a pair of two distinct

cell-types in each organ across three aging-stages. The distribution of the pairwise cWF ratios was calculated for each aging-stage. The ratios of the pairwise cWFs in each cell-type, instead of the cWFs themselves, were used as the cWFs are relative values calculated from the normalized count datasets of the real whole-organ RNA-seq and thus the cWFs values themselves can be used only for their comparison among the cell-types within each organ at each aging-stage (i.e., the cWFs cannot be compared across the aging-stages). Hence, their pairwise ratios were used for the comparisons across the aging-stages. For each pairwise cell-type within an organ, the distribution of the cWF ratios was calculated for all pair combinations of the consisting cells. For example, assuming that cell-type X has 1,000 cells and cell-type Y has 500 cells with one cWF per cell, respectively, the distribution of the ratio of cWFs of cell-type X vs. Y consisting of 500,000 values of the ratios was calculated. When a cell-type contains zero value of cWF, we substituted the zero value for the second minimum cWF value in the cell-type to avoid the division by zero. Next, for each cell-types pair, we performed Mann-Whitney U-test to make statistical evaluations of the differences of their ratio-distributions between the aging-stages (i.e., 18 mos. vs. 3 mos., 24 mos. vs. 3 mos., 24 mos. vs. 18 mos.). Zeros were generated if the resulting values were below the lower bound of R program, 1e-308. These analyses detected their statistically significant ($p < 0.01$ by U-test) changes of many of the cell type-pairs in all organs over the course of aging (Fig 8).

## Machine learning prediction of the aging-stages by the differential cWF ratios

We next investigated whether such differential pairwise ratio-changes of the cWFs predict the progression of aging. For this purpose, we applied a machine learning algorithm, LightGBM [51], to the dataset of the pairwise cWF ratios between the cell type-pairs listed above. For this analysis, we omitted the data of the cell-types pairs of which statistical analysis described above resulted in $p \geq 0.01$ across all pairs of the aging-stages. Using the remaining data, we performed LightGBM classifier analysis [51]. We first generated input data for LightGBM classifier analysis using the cWF ratio distributions obtained as described in the above section. The 100 cWF ratio distribution-values were randomly sampled for each cell-types pair for each aging-stage. The sampled values were at random combined with each other, resulting in the matrix of "the 100 cWFs ratio distribution values" x "the number of the cell-type pairs" for each aging-stage. Using these matrices for all three aging-stages as the input data, we generated a LightGBM classifier of aging-stages on Python, using the '*lightgbm*' package [51]. The input data was divided by 8: 2 into training data and test data. The training data was used for tuning the parameters of LightGBM. The tuning was performed by the '*optuna*' package [52], searching for the best parameters to maximize macro-F1 score. Specifically, the searched parameters are 'max_depth', 'num_leaves', 'subsample', 'subsample_freq', 'colsample_bytree', 'min_child_samples', ranging from 3 to 12, 2 to 256, 0.1 to 1.0, 1 to 7, 0.1 to 1.0, and 5 to 100, respectively. The model was generated using the best parameters and evaluated using test data by accuracy, macro-Precision, macro-Recall, and macro-F1_score. These evaluations were performed by a Python package, '*scikit-learn*' [53] and their formulas are as follows:

$$accuracy = \frac{1}{3} \sum\nolimits_{a=1}^{3} \frac{TP_a + TN_a}{TP_a + FP_a + TN_a + FN_a} \tag{13}$$

$$macro - Precision = \frac{1}{3} \sum\nolimits_{a=1}^{3} \frac{TP_a}{TP_a + FP_a} \tag{14}$$

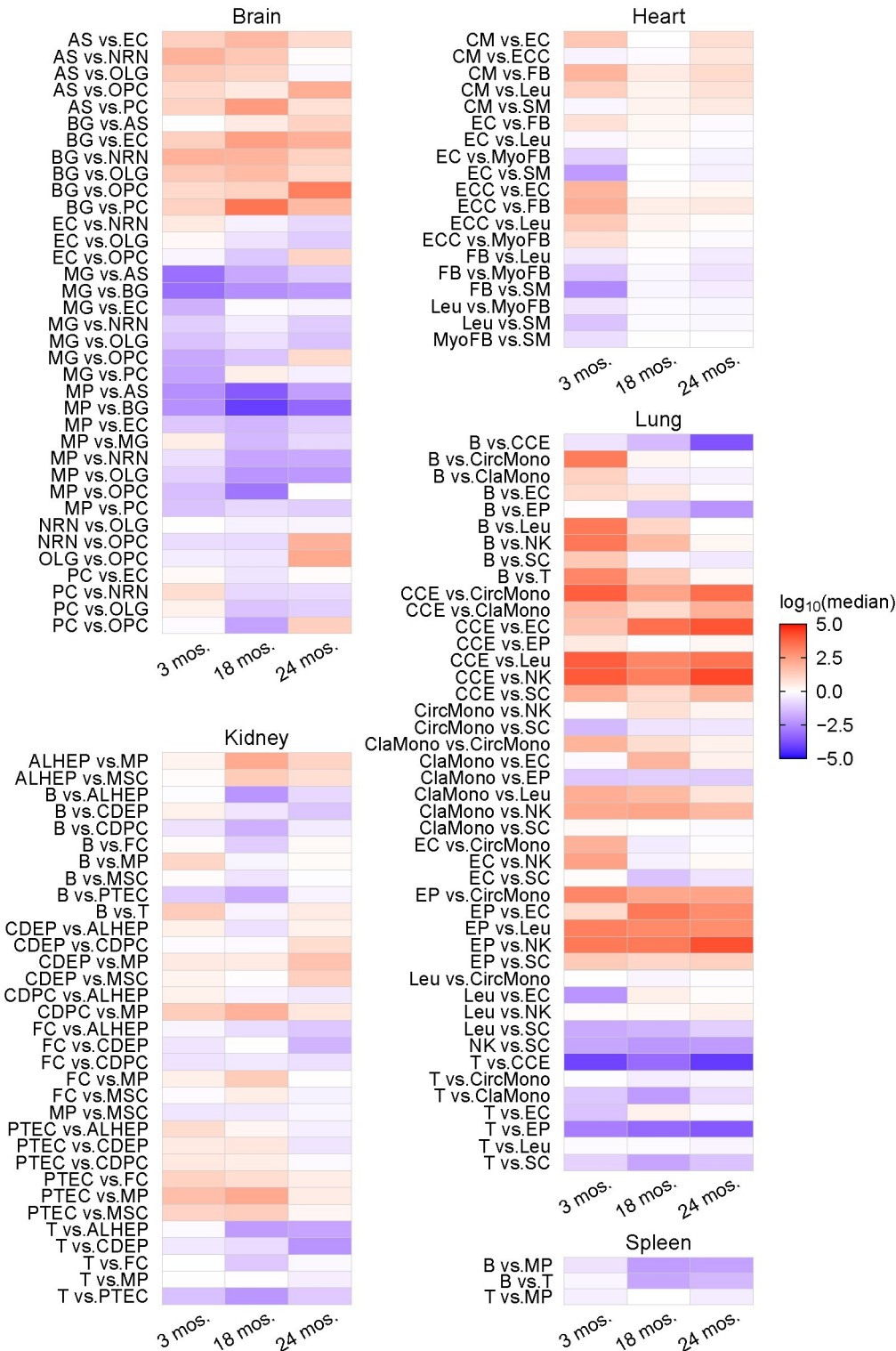

**Fig 8. Differential cWFs during aging of mouse.** The changes of the ratios of cWFs of cell type-pairs across three aging-stages (3 months-old: 3 mos., 18 months-old: 18 mos., 24 months-old: 24 mos.) are shown as heatmap for each organ (Brain, Heart, Kidney, Lung, Spleen). The cell type-pairs are indicated at the left of each heatmap. The ratios are the pairwise ratios of the corresponding cell-types. For example, AS vs. EC indicates the ratio of the cWF for AS (numerator) and EC (denominator). The ratios are indicated as $\log_{10}$ of the medians. Shown are the pairs resulted in p < 0.01 by U-test in one or more of the three pairwise aging stage-comparisons (18 mos. vs. 3 mos., 24 mos. vs. 3

mos., 24 mos. vs. 18 mos.). The analysis was conducted with those showing the significant improvements in the reconstitution (S9 Fig) and the best deconvolution results (S21 Table) for each organ. The full names of the cell type abbreviations and the raw data of the heatmap are found in S22 Table.

$$\text{macro} - \text{Recall} = \frac{1}{3} \sum\nolimits_{a=1}^{3} \frac{TP_a}{TP_a + FN_a} \tag{15}$$

$$\text{macro} - \text{F1score} = \frac{1}{3} \sum\nolimits_{a=1}^{3} \frac{2}{\left(\frac{TP_a}{TP_a + FP_a}\right)^{-1} + \left(\frac{TP_a}{TP_a + FN_a}\right)^{-1}} \tag{16}$$

where, $TP_a$, $FP_a$, $TN_a$ and $FN_a$ represent true positive, false positive, true negative, false negative, respectively, in the classification of the aging stage, $a$. In addition, we also extracted feature importance based on 'gain' from the model and confirmed the high-ranked contributors in the model (i.e., aging-stage classification).

First, each organ dataset was independently evaluated (Fig 9). The result shows high predictability with the pairwise cWF ratios between the brain, the lung, the heart, and the kidney cells, as indicated by their high prediction index scores (i.e., accuracy, macro-recall, macro-precision, macro-F1 score) of 0.88–0.98 (Fig 9A).

The LightGBM prediction with all organs combined also shows high performance as indicated by its prediction index scores of 1 (Fig 10A). The feature importance analysis shows the top-ranked cell type-pairs are those of the lung and the brain, supporting the higher performance results with the cell-types of these organs when they are independently analyzed (Fig 9A). The cell-types indicated as important contributors to the prediction listed above for the individual organ analyses also ranked higher in this all-organ-combined analysis (Fig 10B). The spleen shows significantly lower performance (Fig 9). This may imply that the transcriptome-sizes of three cell-types (B cell, T cell, macrophage) of this organ are similar. Alternatively, the prediction performance is lower due to the small number of cell-types features (i.e., three cell-types). Taken together, the results demonstrate that the differential cWF ratios of the brain, the lung, the heart and the kidney accurately predict the progression of aging, suggesting a possible role of the differential transcriptome-size among the cells in these organs in the aging mechanism. The results also indicate a possibility of using these indices to determine the progression of aging.

## Discussion

In this study, we report a computational method to calculate cWFs, coefficients to offset the lack of the representation of transcriptome size variations and differential cryptic gene expression among different cells of solid organs (Fig 2). Using this method, we computed and describe cWFs for 76 cell-types across 10 organs (Fig 3). We show that they indeed account for the relative variations of the transcriptome-size of the cells and their differential cryptic gene expression by demonstrating their requisite role for accurately reconstituting and deconvolving the whole-organ RNA-seq data using their composite scRNA-seq data weighted by the corresponding cWFs (Figs 4–6). We also show that they recapitulate the experimentally-determined total mRNA content-differences, using PBMC data (Fig 7).

Furthermore, we show that cWFs differentially change among diverse cell-types across various organs over the course of aging and can be used to effectively predict aging progression in the aging mouse model (Figs 8–10). These results suggest a possible role of differential

**A**

|        | Accuracy | macro-Precision | macro-Recall | macro-F1 Score |
|--------|----------|-----------------|--------------|----------------|
| Brain  | 0.983    | 0.984           | 0.983        | 0.983          |
| Heart  | 0.883    | 0.893           | 0.883        | 0.882          |
| Kidney | 0.883    | 0.883           | 0.883        | 0.881          |
| Lung   | 0.967    | 0.967           | 0.967        | 0.967          |
| Spleen | 0.517    | 0.514           | 0.517        | 0.514          |

**B**

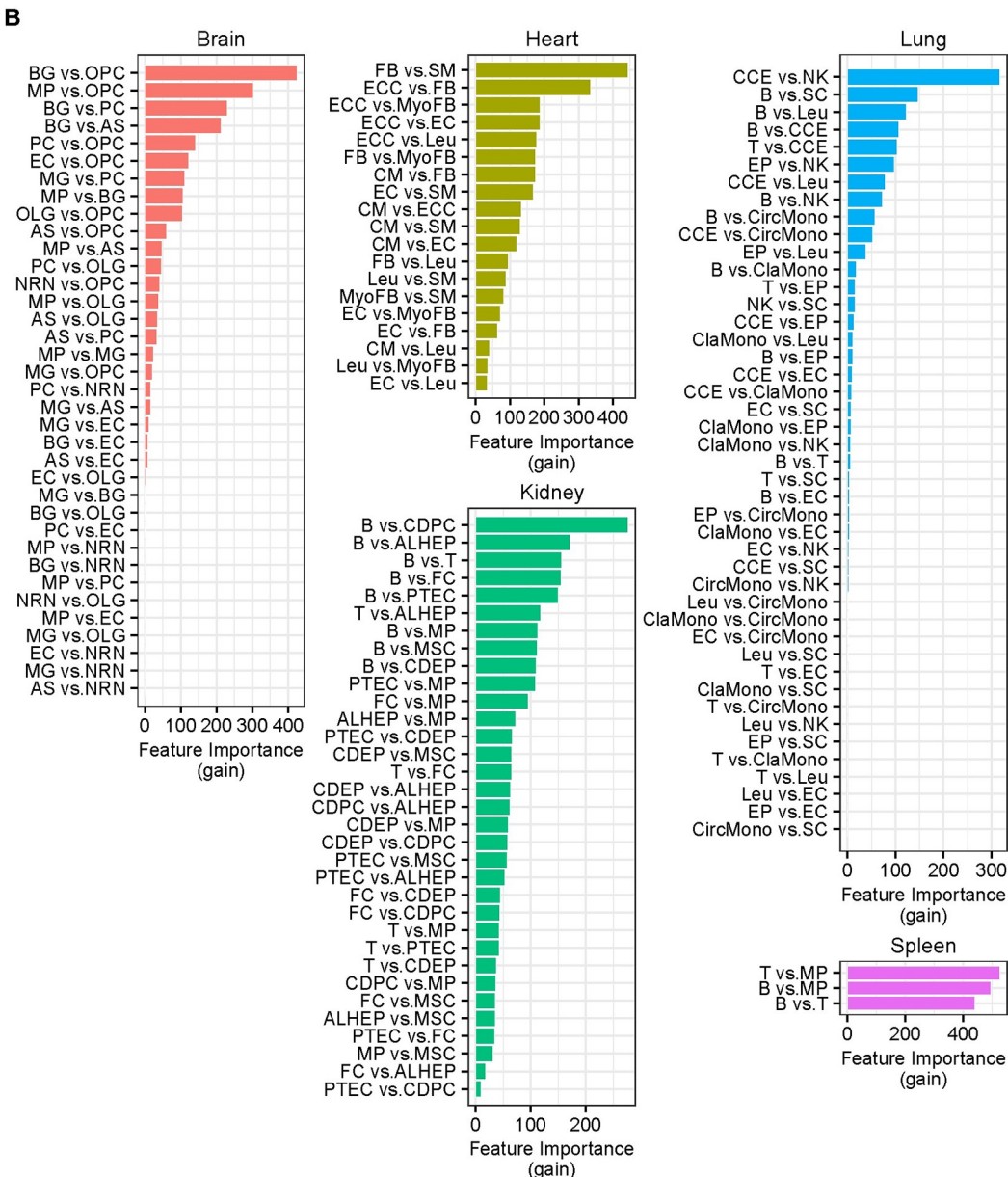

**Fig 9. Prediction of the aging-stage by the differential cWFs of each organ.** **(A)** The prediction scores (accuracy, macro-precision, macro-recall, macro-F1 score) for each organ (Brain, Heart, Kidney, Lung, Spleen) are shown. **(B)** The top-ranked features (i.e., pairwise cell-types) important for the prediction are shown as bar graphs for each organ. The feature importance is indicated by gain. The corresponding raw data are found in Sheet A in S23 Table.

**A**

| | Accuracy | macro-Precision | macro-Recall | macro-F1 Score |
|---|---|---|---|---|
| 5 organs | 1.000 | 1.000 | 1.000 | 1.000 |

**B**

**Fig 10. Prediction of the aging-stage by the differential cWFs of all 5 organs combined. (A)** The prediction scores (accuracy, macro-precision, macro-recall, macro-F1) by combining the cWF of all pairwise cell-types across all organs are shown. **(B)** The top 30 important features (i.e., the pairwise ratios of the cWFs) for the prediction are shown as bar graphs for each organ. The feature importance is indicated by gain. The cell type-pairs and their organs are indicated as organ-name_cell-type pair (e.g., Lung_CCE vs. NK: the CCE vs. NK pair of the lung). The corresponding raw data are found in Sheet B in S23 Table.

transcriptome size-change in the aging mechanism. The results also indicate their possible diagnostic applications as biomarkers for aging progression and/or aging associated diseases.

Despite the usefulness of cWFs, some limitations must be noted. While our method is effective for the organs consisting of heterogeneous cell-types, it is ineffective for a relatively homogenous organ such as the liver consisting of predominantly hepatocytes (>95%) (Figs 3, 5, S4, and S5). Another limitation could be the requirement of both bulk and single-cell RNA-seq data of the organs and the tissues of interest. Our method requires both of these datasets for the target organs/tissues to infer the cWFs of the composing cell-types. However, this may become less of a limiting factor as the availability of such data is exponentially growing. Once we determine the cWFs for the specific cell-types of organs, the same cWFs can be used for the same cell-types of the corresponding organs derived from other independently-generated

datasets (i.e., regardless of their sources). Therefore, the cWFs reported in this study can be used for the same 76 cell-types and 10 organs for other studies. Another requirement is that the method needs at least a rough a priori knowledge about the cell-type compositions (i.e., the ratio of cell-types) of the organs and the tissue of interest –i.e., the method works only in a supervised framework. This limitation could also become less limiting as a growing body of scRNA-seq data provides information on the cell-type compositions of previously less characterized organs/tissues. Furthermore, we show at least one example where the expected pathological changes of the cell-type ratio of the heart can be predicted using the cWFs derived from the healthy control heart (S8 Fig), despite a bias introduced by this reference data.

## Supporting information

**S1 Fig. Sensitivity analysis for the total number of cell subjects, *m*.** The cWFs for each cell type of each organ are shown for the *m* = 50 and *m* = 200. The results are shown for the signature genes numbers (Signature genes#), 100, 300, and 500. Compare the results to those with the default *m* = 100 (Fig 3). Raw data are available as S3 Table.
(TIF)

**S2 Fig. Reconstitution of the whole organ RNA-seq by the composite scRNA-seq and cWFs computed with varying numbers of cell subjects, *m*.** The similarity is shown as violin plot of the Pearson correlation coefficient for each number of the signature genes (100, 300, 500) for each organ (indicated above each plot). The number of *m* is indicated at the bottom of each graph. The results with the default *m* = 100 are the same ones shown in Fig 4. Raw data are available as S6 Table. SkMuscle: skeletal muscle.
(TIF)

**S3 Fig. Accurate reconstitution of the whole organ RNA-seq by the composite scRNA-seq and cWFs using two independent RNA-seq datasets.** The Pearson correlation coefficients with and without cWFs are shown for each organ using Quant 3’ mRNA-seq **(A)** (raw data are available as S7 Table) and deep RNA-seq **(B)** (raw data are available as S8 Table). Independently prepared RNA samples are used for each RNA-seq methods.
(TIF)

**S4 Fig. cWFs computed using Quant 3’ mRNA-seq data.** Shown are cWFs for each cell-type of each organ. Compare the results to those shown in Fig 3. Raw data are available as S9 Table.
(TIF)

**S5 Fig. cWFs computed using deep RNA-seq data.** Shown are cWFs for each cell-type of each organ. The sequencing method is the same as those of Fig 3; however, the organ/RNA samples are independently prepared and sequenced. Compare the results to those shown in Fig 3. Raw data are available as S10 Table.
(TIF)

**S6 Fig. Reconstitution and deconvolution of the bulk human PBMC RNA-seq by the composite scRNA-seq.** **(A)** Reconstitution results with and without (no) cWFs are compared. The results with the 100, 300, 500 signature genes are shown. The similarity is shown as violin plot of the Pearson correlation coefficients. The corresponding raw data with no cWFs and with cWFs are found in S14 and S15 Tables, respectively. **(B)** Bar graph showing the cell type-ratios computed by the deconvolution method (V-scRNAseq) for each organ. The deconvolution was performed with the cWFs computed using the optimal number of the signature genes for each organ (indicated in the accompanying S16 Table, where the best performing result–i.e., the lowest RMSE and the highest Pearson correlation coefficient shown in S6B Fig is

highlighted in light green). The bar graphs are composed of the cell-types computed to be present for each organ by our method. The similarity scores (RMSE: Root Mean Squared Errors, Pearson correlation coefficient) are indicated at the top of the bar.
(TIF)

**S7 Fig. Accurate representation of the experimentally known differential cellular RNA contents between HEK293T and Jurkat cells regardless of their differential ratios.** The cWFs are shown for HEK293T and Jurkat cells when they are mixed at 20: 80 and 80: 20 ratios. The results are shown by both box plots (left) and bar graphs (right). The bar graphs are indicated by mean ± S.E. The results with the signature genes number (Signature genes#), 100, 300, and 500 are shown. Raw data are available as S17 Table.
(TIF)

**S8 Fig. Prediction of putative changes of cell-type ratios of the heart during myocardial infarction (MI).** The ratios for each cell-type (CM: cardiomyocyte, EC: endotheial cell, ECC: endocardial cell, FB: fibroblast, Leu: leukocyte, MyoFB: myofibroblast) at each MI stage (E: early MI, M: middle fibrosis, L: late remodeling) are indicated as their relative fold changes to their corresponding sham controls. The MI stages (E, M, L) are defined as previously described [26]. The cWFs were calculated from the sham operated mice data at each stage, using the number of the signature genes, 300, which was optimal in the analyses of deep RNA-seq data from 11 weeks-old male C57BL6/N Jcl mice. These cWFs of the sham operated mice data are used for the deconvolution of the MI data. The bar graphs are shown as mean ± S.E. Raw data are available as S18 Table.
(TIF)

**S9 Fig. Reconstitution of the whole organ RNA-seq of the aging model mouse by the composite scRNA-seq.** The results with and without (no) cWFs are compared for each aging-stage (3 mos., 18 mos., 24 mos.) for each number of the signature genes (100, 300, 500) and for each organ (Brain, Heart, Kidney, Lung, Spleen). The similarity is shown as violin plots of the Pearson correlation coefficients. The corresponding raw data with no cWFs and with cWFs are found in S19 and S20 Tables, respectively.
(TIF)

**S1 Table. Raw data for Fig 1.**
(XLSX)

**S2 Table. Raw data for Fig 3.** The abbreviations of cell types (Sheet A in S2 Table) and the raw data for Fig 3 (Sheet B in S2 Table) are shown.
(XLSX)

**S3 Table. Raw data for S1 Fig.**
(XLSX)

**S4 Table. Summary of datasets used in this paper.**
(XLSX)

**S5 Table. Raw data for Fig 4.**
(XLSX)

**S6 Table. Raw data for S2 Fig.**
(XLSX)

**S7 Table. Raw data for S3A Fig.**
(XLSX)

**S8 Table. Raw data for S3B Fig.**
(XLSX)

**S9 Table. Raw data for S4 Fig.**
(XLSX)

**S10 Table. Raw data for S5 Fig.**
(XLSX)

**S11 Table. Raw data for Fig 5B and 5C.** The raw data for Fig 5B (Sheet A in S11 Table) and 5C (Sheet B in S11 Table) are shown.
(XLSX)

**S12 Table. Raw data for Fig 6.** The raw data for Fig 6 (Sheet A in S12 Table) and the mean counts of the "Virtual scRNAseq" (Sheet B in S12 Table) are shown.
(XLSX)

**S13 Table. Raw data for Fig 8.**
(XLSX)

**S14 Table. Raw data for S6A Fig with no cWFs.**
(XLSX)

**S15 Table. Raw data for S6A Fig with cWFs.**
(XLSX)

**S16 Table. The optimal number of the signature genes for each organ shown in S6 Fig.** The deconvolution results for each signature gene numbers (100, 300, 500) and the reference cell type ratios of human PBMC are shown in Sheet A in S16 Table. The abbreviations of cell types are described in Sheet B in S16 Table.
(XLSX)

**S17 Table. Raw data for S7 Fig.**
(XLSX)

**S18 Table. Raw data for S8 Fig.** The fold-changes (FC) of each cell type in the MI vs. sham hearts (Sheet A in S18 Table) and their means (Sheet B in S18 Table) are shown.
(XLSX)

**S19 Table. Raw data for S9 Fig with no cWFs.**
(XLSX)

**S20 Table. Raw data for S9 Fig with cWFs.**
(XLSX)

**S21 Table. The deconvolution results using *Tabula Muris Senis* datasets.** The best results (i.e., the lowest RMSE and the highest Pearson correlation coefficients) are highlighted in light-green. The failed improvements with cWFs are highlighted in pink.
(XLSX)

**S22 Table. The full names of the cell type abbreviations and the raw data for the heatmap shown in Fig 8.** The abbreviations of cell types (Sheet A in S22 Table) and the raw data for Fig 8 (Sheet B in S22 Table) are shown.
(XLSX)

**S23 Table. Raw data for Figs 9B and 10B.** The raw data for Fig 9B (Sheet A in S23 Table) and 10B (Sheet B in S23 Table) are shown.
(XLSX)

# Acknowledgments

We thank K. Sugisaka, R. Takahashi, T. Ninomiya, R. Kitaura, R. Ishikawa, S. Taniyama for their administrative and laboratory management assistance. We are also grateful to the members of Karydo TherapeutiX, Inc. and Sato laboratory at ATR for advice and discussion throughout the course of this work.

# Author Contributions

**Conceptualization:** Thomas N. Sato.

**Data curation:** Kengo Tejima, Satoshi Kozawa.

**Formal analysis:** Kengo Tejima, Satoshi Kozawa, Thomas N. Sato.

**Funding acquisition:** Thomas N. Sato.

**Investigation:** Kengo Tejima, Satoshi Kozawa, Thomas N. Sato.

**Methodology:** Kengo Tejima, Satoshi Kozawa, Thomas N. Sato.

**Project administration:** Thomas N. Sato.

**Supervision:** Thomas N. Sato.

**Validation:** Kengo Tejima, Satoshi Kozawa, Thomas N. Sato.

**Writing – original draft:** Kengo Tejima, Satoshi Kozawa, Thomas N. Sato.

**Writing – review & editing:** Kengo Tejima, Satoshi Kozawa, Thomas N. Sato.

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
