## [Decision Letter · Decision Letter 0]

7 Mar 2024

Dear Dr Sato,

Thank you very much for submitting your Methods entitled 'Cell type-specific weighting-factors to solve solid organs-specific limitations of single cell RNA-sequencing' to PLOS Genetics.

The manuscript was fully evaluated at the editorial level and by independent peer reviewers. The reviewers appreciated the attention to an important problem, but raised some substantial concerns about the current manuscript. Based on the reviews, we will not be able to accept this version of the manuscript, but we would be willing to review a much-revised version. We cannot, of course, promise publication at that time.

If you decide to revise the manuscript for further consideration at PLOS Genetics, please aim to resubmit within the next 60 days, unless it will take extra time to address the concerns of the reviewers, in which case we would appreciate an expected resubmission date by email to plosgenetics@plos.org.

We are sorry that we cannot be more positive about your manuscript at this stage. Please do not hesitate to contact us if you have any concerns or questions.

Yours sincerely,

Mingyao Li

Academic Editor

PLOS Genetics

Michael Epstein

Section Editor

PLOS Genetics

Reviewer's Responses to Questions

**Comments to the Authors:**

Reviewer #1: In this manuscript, the authors introduce a novel metric called the "cell-type-specific weighting-factor (cWF)" to address a limitation in current single-cell RNA-seq technology. The internal normalization process in the analyses currently used tends to eliminate putative transcriptome size variations among different cells, thereby obscuring potentially significant functional differences. The literature has already highlighted the importance of transcriptome size in cellular analyses. The authors validated the proposed metric through the reconstitution and deconvolution of bulk RNA-seq data and demonstrated its predictive capability for aging progression.

However, I still have some comments below hope to be addressed.

Major Comments:

1. The validation of the cWF involves re-construction of the Bulk RNA-seq data from both sc-RNA seq data and estimated cWF (from Figure 1 and Figure 4). The substantial improvements have been observed. However, could the author please clarify whether the estimated Pearson correlation for Figure 4 is from the results of within-sample prediction or out-of-sample prediction? If the results are from within-sample prediction, could the author provide the results from out-of-sample prediction? Would this be a potential way of addressing the limitation of the current method, that is requiring both sc-RNAseq and Bulk RNAseq at the same time? It will be very helpful to set up more discussions.

2. Page 9, 162. The reconstruction of the Buk RNA-seq data involved the mean value of cWF for each cell type. However, the distributions of cWF under certain cell types and cancer types are highly skewed and with larger variation, could the author provide more rational os using the mean value of cWF? Could the author conduct some sensitivity analysis to evaluate results using other robust measures, i.e. median?

3. Page 20, 376, could the author please have more description of the definition of the prediction accuracy? For Figure 9A, could the author provide more discussion of why the prediction accuracy for Spleen is way lower than the rest of the cancer types?

4. Page 27, 569, the total number of cell subjects, m, is arbitrarily set to 100. Would the estimated cWF vary with a different number of m selected? More sensitive analysis is recommended to see how robust the estimation of cWF w.r.t to the total number of cell subjects, m.

Minor Comments:

1. Page 12, 194, there should be space between “t”and “=”.

2. Page 29, 616, would zero-truncated normal help instead of setting zeros for negative value?

Reviewer #2: The manuscript titled "Cell type-specific weighting-factors to solve solid organs-specific limitations of single cell RNA-sequencing" introduces cell type-specific weighting-factors (cWFs) to solve the critical limitations of transcription size variability during scRNA-seq analysis. And the authors also performed cWFs to effectively predict aging progression, suggesting the diagnostic applications and association with aging mechanisms. While the concept of cWF is interesting, there are potential limitations in the calculation especially the challenges in distinguishing between the cWF and the cell type population. Detailed comments bellow:

1. The weight for transcription size is calculated by solving quadratic problem, which has led to difficulties in distinguishing the weight from the real population percentage for the samples without cell-type ratios. In addition, Since the transcription size for a specific cell type should ideally remain consistent across different conditions, I am wondering if a specific cell type is sharing consistent cWF among different sequencing samples?

2. It is important to investigate whether the cell type proportion affects the weight of transcription size. In other words, if the authors simulate bulk RNA-seq data with different compositions of the same cell types, is the cWF changing along with the proportion changes? Furthermore, if the authors use different weights to simulate bulk RNA-seq data, it is essential to assess whether the cWFs algorithm can accurately predict the weight.

3. In Fig 3, it is observed that cWF values for the majority of cell types are around 0, indicating low transcription size. According to the formula in Fig 2, which states that the transcription size equals scRNA-seq times cWF, this suggests that the majority of these cell types did not express any RNA molecule, which is biologically unreasonable.

4. It is indicating the cWF calculation require the input of ‘celltype_ref_ratio’ in the document of github. Then, the requirement for 'celltype_ref_ratio' in the calculation of cWF would introduce bias in the deconvolution process, as the authors utilize the real ratio as the input.

5. The authors utilized cWF for deconvolution in Fig 5. As shown in Fig 3, the SEC in the liver exhibits a higher cWF value than other cell types, indicating that the SEC has a larger transcription size and would reduce the percentage of SEC during deconvolution. However, when compared with the true percentage, DWLS and MuSiC results, the SEC cell type composition increases in cWF. This discrepancy between the cWF and deconvolution results presents a controversial aspect.

**Have all data underlying the figures and results presented in the manuscript been provided?**

Reviewer #1: Yes

Reviewer #2: Yes

PLOS authors have the option to publish the peer review history of their article (what does this mean?). If published, this will include your full peer review and any attached files.

Reviewer #1: No

Reviewer #2: No

---

## [Decision Letter · Decision Letter 1]

8 Jul 2024

Dear Dr Sato,

Thank you very much for submitting your Methods entitled 'Cell type-specific weighting-factors to solve solid organs-specific limitations of single cell RNA-sequencing' to PLOS Genetics.

The manuscript was fully evaluated at the editorial level and by independent peer reviewers. While Reviewer 2 was satisfied with the revision, Reviewer 1 had one additional comment related to the sensitivity analysis in the revision that requires a response. We ask you to address this comment in a revised manuscript.

Yours sincerely,

Mingyao Li

Academic Editor

PLOS Genetics

Michael Epstein

Section Editor

PLOS Genetics

Reviewer's Responses to Questions

**Comments to the Authors:**

Reviewer #1: Thank you for addressing my comments comprehensively. However, I have one additional comment that requires your attention.

Major Comments:

1. In response to the previous major comment 4, you have included a sensitivity analysis to evaluate the estimation of cWF with respect to different numbers of cell subjects, m. However, in Figure S1, the magnitude of the estimated cWF varies significantly with different choices of m, even under the same cancer type and with the same number of signature genes. Could you please provide a more detailed explanation for this variation? Additionally, how might this variation affect the accuracy of reconstituting the whole organ RNA-seq?

Reviewer #2: The authors have addressed all comments and questions effectively, and the cWF has proven to be a great tool for evaluating transcription size for different cell types. I recommend accepting this paper as it is suitable for publication in PLOS Genetics.

**Have all data underlying the figures and results presented in the manuscript been provided?**

Reviewer #1: Yes

Reviewer #2: Yes

PLOS authors have the option to publish the peer review history of their article (what does this mean?). If published, this will include your full peer review and any attached files.

Reviewer #1: No

Reviewer #2: No

---

## [Decision Letter · Decision Letter 2]

20 Sep 2024

Dear Dr Sato,

We are pleased to inform you that your manuscript entitled "Cell type-specific weighting-factors to solve solid organs-specific limitations of single cell RNA-sequencing" has been editorially accepted for publication in PLOS Genetics. Congratulations!

Yours sincerely,

Mingyao Li

Academic Editor

PLOS Genetics

Michael Epstein

Section Editor

PLOS Genetics

Comments from the reviewers (if applicable):

Reviewer's Responses to Questions

**Comments to the Authors:**

Reviewer #1: Thanks for addressing all my concerns.

Reviewer #2: The authors have addressed all my concerns, and I recommend acceptance in PLOS Genetics.

**Have all data underlying the figures and results presented in the manuscript been provided?**

Reviewer #1: None

Reviewer #2: Yes

PLOS authors have the option to publish the peer review history of their article (what does this mean?). If published, this will include your full peer review and any attached files.

Reviewer #1: No

Reviewer #2: No

**Data Deposition**

http://datadryad.org/submit?journalID=pgenetics&manu=PGENETICS-D-24-00035R2

**Press Queries**

---

## [Editor Report · Acceptance letter]

17 Oct 2024

PGENETICS-D-24-00035R2 

Cell type-specific weighting-factors to solve solid organs-specific limitations of single cell RNA-sequencing 

Dear Dr Sato, 

We are pleased to inform you that your manuscript entitled "Cell type-specific weighting-factors to solve solid organs-specific limitations of single cell RNA-sequencing" has been formally accepted for publication in PLOS Genetics! Your manuscript is now with our production department and you will be notified of the publication date in due course.

With kind regards,

Lilla Horvath

PLOS Genetics

On behalf of:
